# Holocene seasonal temperature evolution and spatial variability over the Northern Hemisphere landmass

**Wenchao Zhang** [1,2], **Haibin Wu** [1,3,4] ✉, **Jun Cheng** [5] ✉, **Junyan Geng** [1,4], **Qin Li** [1,6], **Yong Sun** [7], **Yanyan Yu** [1], **Huayu Lu** [8] & **Zhengtang Guo** [1,3,4]

The origin of the temperature divergence between Holocene proxy reconstructions and model simulations remains controversial, but it possibly results from potential biases in the seasonality of reconstructions or in the climate sensitivity of models. Here we present an extensive dataset of Holocene seasonal temperatures reconstructed using 1310 pollen records covering the Northern Hemisphere landmass. Our results indicate that both summer and winter temperatures warmed from the early to mid-Holocene (~11–7 ka BP) and then cooled thereafter, but with significant spatial variability. Strong early Holocene warming trend occurred mainly in Europe, eastern North America and northern Asia, which can be generally captured by model simulations and is likely associated with the retreat of continental ice sheets. The subsequent cooling trend is pervasively recorded except for northern Asia and southeastern North America, which may reflect the cross-seasonal impact of the decreasing summer insolation through climatic feedbacks, but the cooling in winter season is not well reproduced by climate models. Our results challenge the proposal that seasonal biases in proxies are the main origin of model–data discrepancies and highlight the critical impact of insolation and associated feedbacks on temperature changes, which warrant closer attention in future climate modelling.

Knowledge of past climatic conditions is essential to improve our understanding of the climate system with respect to the climate forcings and feedbacks, and to better constrain future climate projection[1]. However, the long-term evolution of global temperatures over the past 11 kyr (the Holocene epoch) remains poorly constrained. Multiproxy reconstructions[2–4] (MC13 and KA20, Fig. 1a) indicate an early to mid-Holocene Thermal Maximum (HTM), highlighting the crucial effect of boreal summer insolation on global temperature change[2]. By contrast, climate models simulate a long-term warming trend throughout the Holocene associated with ice-sheet retreat and rising greenhouse gas concentrations[5–8]. This Holocene temperature conundrum was suggested to result from the poor spatial representation of proxy records[8] or seasonal biases in proxies[5–7], since most of them might represent warm-season temperatures according to studies focusing on oceans[6]. Robust seasonal, especially winter, temperature reconstructions over vast land areas can provide a new and crucial perspective on this conundrum[9–11].

The HTM in summer season has been found across many regions of Europe[12–14], Asia[15], and North America[16], as is shown in multiproxy stacks[3,4] (KA20, Fig. 1b), although long-term warming trends have been inferred for southern Europe[12,14] and west-central North America[16]. However, the winter temperature trend during the Holocene is poorly documented and intensely debated. A long-term winter warming trend was inferred at individual sites in Eurasia[10,11], Alaska[17], and in a continental stack for North America and Europe[9] (MA18, Fig. 1c), but it conflicts with other lines of evidence suggesting winter cooling from

**Fig. 1 | Multiproxy records of annual and seasonal temperature evolution during the Holocene. a** Annual temperature trends from our pollen-based reconstructions over the Northern Hemisphere (NH) landmass (red), North America and Europe (NAEU, teal), and Asia (dark blue) with the number of 2° × 2° grid cells for the temperature stack in 200-yr bins (grey bar) and mean squared chord distance (SCD; blue). This panel also shows a marine-dominated multiproxy stack for 30–90°N[2] (MC13), NH marine and terrestrial multiproxy stack[3] (KA20), and pollen-based stack for North America and Europe[9] (MA18). **b** Summer temperature trends from Kaufman et al.[3] (KA20), our pollen-based reconstructions over the NH landmass (red) and in North America and Europe (NAEU, teal), and GDD₅ record from Marsicek et al.[9] (MA18). **c** Winter temperature trends from Kaufman et al.[3] (KA20), Marsicek et al.[9] (MA18), and our reconstruction (red). Shading indicates 95% uncertainty bands for all curves except for MC13 (1σ uncertainty). KA20s are composite z-score curves averaged from the original zonal stacks.

the mid-Holocene onwards in Europe[14], East Asia[18], North America[19], and in Northern Hemisphere (NH) multiproxy stacks[3,4] (KA20, Fig. 1c). Discrepancies among records may be due to the spatial heterogeneity of climate change[8,20], or to inconsistencies among proxies. Therefore, it is important to use reliable records with a large spatial coverage and fine geographic representation to comprehensively understand Holocene climate change[8].

Fossil pollen reflects past vegetation communities that have specific temperature requirements in both cold and warm seasons[9,21–23], and thus it is a potentially reliable proxy for seasonal temperature reconstructions[9] except in arid regions where water availability may be a more constraining factor. Here we use extensive pollen records from the NH landmass (Supplementary Fig. 1) to quantitatively reconstruct Holocene seasonal and annual temperatures using the modern

analogue technique (MAT) based on plant functional type (PFT) scores (Methods). We compare our data with a transient climate simulation (TraCE-21ka) from the Community Climate System Model 3[5] (CCSM3; Methods) in hemispheric and regional scales. Our extensive dataset indicates evident cooling trends in both summer and winter seasons after ~7 ka BP over the NH landmass, which suggest a low risk of proxy origin of Holocene temperature conundrum and provide insights into the mechanisms of Holocene temperature change.

## Results
### Seasonal temperature evolution
Our results show that the trends of NH annual, summer and winter temperatures are all characterised by a rapid warming from the early to mid-Holocene (11–7 ka BP) and gradual cooling after ~7 ka BP (Fig. 1).

This indicates that the mid-Holocene was the warmest interval of the Holocene at the hemispheric scale. In terms of amplitude, the early to mid-Holocene warming was much greater in winter (~4.1 °C) than in summer (~1.4 °C), but the cooling after ~7 ka BP was comparable between winter (~0.7 °C) and summer (~0.5 °C). Consequently, the seasonality of temperature (i.e., the difference between summer and winter temperatures) shows a long-term decrease (-0.22 °C ka$^{-1}$), which was most pronounced during the early to mid-Holocene (-0.58 °C ka$^{-1}$; see below).

The annual warming trend from the early to mid-Holocene accords with another pollen-based terrestrial stack for North America and Europe[9] (MA18), and NH multiproxy reconstructions[3,4] (KA20), while another multiproxy stack[2] (MC13) exhibits an HTM phase (Fig. 1a). The long-term cooling trend after ~7 ka BP has been widely detected in syntheses of proxy records[2–4,24–26], in global/NH ocean temperature records[2,3,8], and accords with the general pattern of NH glacier advances[27], but it is not well shown in the previous pollen-based reconstruction for North America and Europe[9] (MA18, Fig. 1a), and in a model–data assimilation output of global surface temperature constrained by marine proxy records[8].

Regarding seasonal temperatures, the timing of the mid-Holocene HTM is in good agreement with the multiproxy records from Kaufman et al.[3] for both summer and winter seasons (KA20, Fig. 1b, c), and also generally consistent with a summer-related record of growing degree days above 5 °C (GDD$_5$) from North America and Europe[9] (MA18, Fig. 1b). For the winter temperature, the warming trend from ~11 to 7 ka BP is in accord with reconstructions of Marsicek et al.[9], but the subsequent cooling trend is opposite to them, which demonstrated a continuous warming until ~2 ka BP in North America and Europe (Fig. 1c). We propose (and demonstrate below) that the divergence among these various stacks is caused mainly by a combination of the spatial variability of climate change and differences in site distributions, although uncertainties associated with the different proxies and methods used may also contribute.

## Spatial variability of temperature change

To elucidate the spatial features of temperature variability, we used empirical orthogonal function (EOF) analysis of selected records spanning the 11–7 ka BP and 7–0 ka BP periods. The first principal components (PC1) show similar early to mid-Holocene rising (Fig. 2c) and mid-to-late Holocene falling trends (Fig. 2d) among the annual, summer and winter temperatures, and the corresponding EOF1 displays well-defined spatial patterns (Fig. 2a, b). According to the EOF1 patterns and geography, we defined eight regions to further clarify the spatial variability of temperature evolution (Fig. 2 and Supplementary Fig. 4). In general, the early to mid-Holocene warming trends were located mainly in eastern North America, Europe, and northern Asia, whereas the cooling trends after ~7 ka BP were widespread over the NH landmass, except for long-term warming in the southern North America and northern Asia. The EOF1 patterns and regional stacks also show evident differences between seasons. In western North America, summer temperatures exhibit a mid-Holocene maximum, which is opposite to the winter season. A long-term summer cooling since the early Holocene occurred in western and southeastern Europe, which contrasts with the long-term warming trend in western Europe and the mid-Holocene HTM in southeastern Europe in the winter season.

Our spatial patterns of annual and seasonal temperature variability generally agree with previous reconstructions[2,9,14] and our independent proxy compilations (see Methods), as indicated by the EOF results for the 11–7 ka BP (Supplementary Fig. 5) and 7–0 ka BP (Supplementary Fig. 6) periods, and the regional stack curves (Supplementary Fig. 4). For the annual temperatures, we attribute the divergence of early to mid-Holocene trends between Marcott et al.[2] and our stack (Fig. 1a) mainly to their low coverage over land where the warming was widely detected, especially in eastern North America and

Europe (Supplementary Figs. 4 and 5), which was also highlighted by Marsicek et al.[9].

The divergence of annual temperature trends between our results and Marsicek et al.[9] in North America and Europe (Fig. 1a) is mainly derived from the difference in western North America, since similar trends were found in eastern North America and Europe (Supplementary Figs. 4–6). In western North America, our much more records can provide a comprehensive view, and indicate an HTM pattern, rather than a long-term warming trend[9] (Supplementary Figs. 4–6). Therefore, the nearly undetected cooling trend after ~7 ka BP from Marsicek et al.[9] (Fig. 1a) may result from the influence of the strong warming trends in southeastern North America, considering their scarcity of records from western North America, together with the absence of data from Asia (Supplementary Figs. 4 and 6).

A similar situation is evident for the winter season. EOF1 patterns and regional stacks from eastern North America and Europe are generally consistent between our results and Marsicek et al.[9] (Supplementary Figs. 4–6), which are also consistent with another pollen-based reconstruction from Europe[14] (Supplementary Fig. 4). With the denser distribution of records in western North America and the new data from Asia, the winter temperature over the NH landmass developed a cooling trend after ~7 ka BP, which contrasts with the long-term warming trend until ~2 ka BP, suggested by Marsicek et al.[9] (Fig. 1c). In addition, several qualitative records based on the $\delta^{18}$O of stalagmites and ice-wedges suggest a long-term winter warming trend during the Holocene in Eurasia[10,11]. This warming trend is supported by our pollen-based reconstructions from nearby sites (Supplementary Fig. 7) and the northern Asia stack (Supplementary Fig. 4). Therefore, a long-term Holocene winter warming trend did occur in some regions, such as northern Asia and southeastern North America, but an overall terrestrial cooling trend dominated the NH landmass after the mid-Holocene.

Although there is an overall agreement in Holocene summer temperature trend[3,4,9,14] (Fig. 1b), some divergence exists between the reconstructions using chironomid and pollen data during the early Holocene. That is, the former tends to show a warmer early Holocene than the latter in northeastern Europe, western North America and northern Asia (Supplementary Fig. 4). The divergence could be partly attributed to spatial inconsistencies, since the EOF1 patterns are overall consistent, but the chironomid records have much lower site density in the mid-latitudes and are dominated by the cooling trend at the high latitudes (Supplementary Fig. 5). Proxy uncertainties may also be an important source of the discrepancy[28–30]. The pollen-based early Holocene temperature may be underestimated due to the delayed tree growth after the long-distance migration to newly glacial retrieved areas[28], while chironomids may be influenced by some other factors than temperature, such as water depth and nutrient availability[29,30].

A local-scale divergence was ever found in southeastern Europe, namely a long-term year-round warming[14] versus cooling[9] throughout the Holocene (Supplementary Fig. 4). Our reconstructions capture an evident seasonal difference, that is, a warmer-than-present summer but a cooler-than-present winter during the early Holocene, and agree better with the independent records[31–33] (Supplementary Fig. 4).

## Model–data comparisons

Based on our extensive seasonal reconstructions across the NH landmass, we made a detailed comparison with the TraCE-21ka simulation results, focusing on seasonal and spatial aspects. For summer temperatures, the simulation output shows that following a rapid rise during the earliest Holocene, a hypsithermal was maintained between ~10 and 7 ka BP, succeeded by long-term cooling (Fig. 3b). This trend is generally consistent with our pollen-based reconstructions (Fig. 3b), although some divergence occurs in northern Asia (Supplementary Figs. 8 and 9).

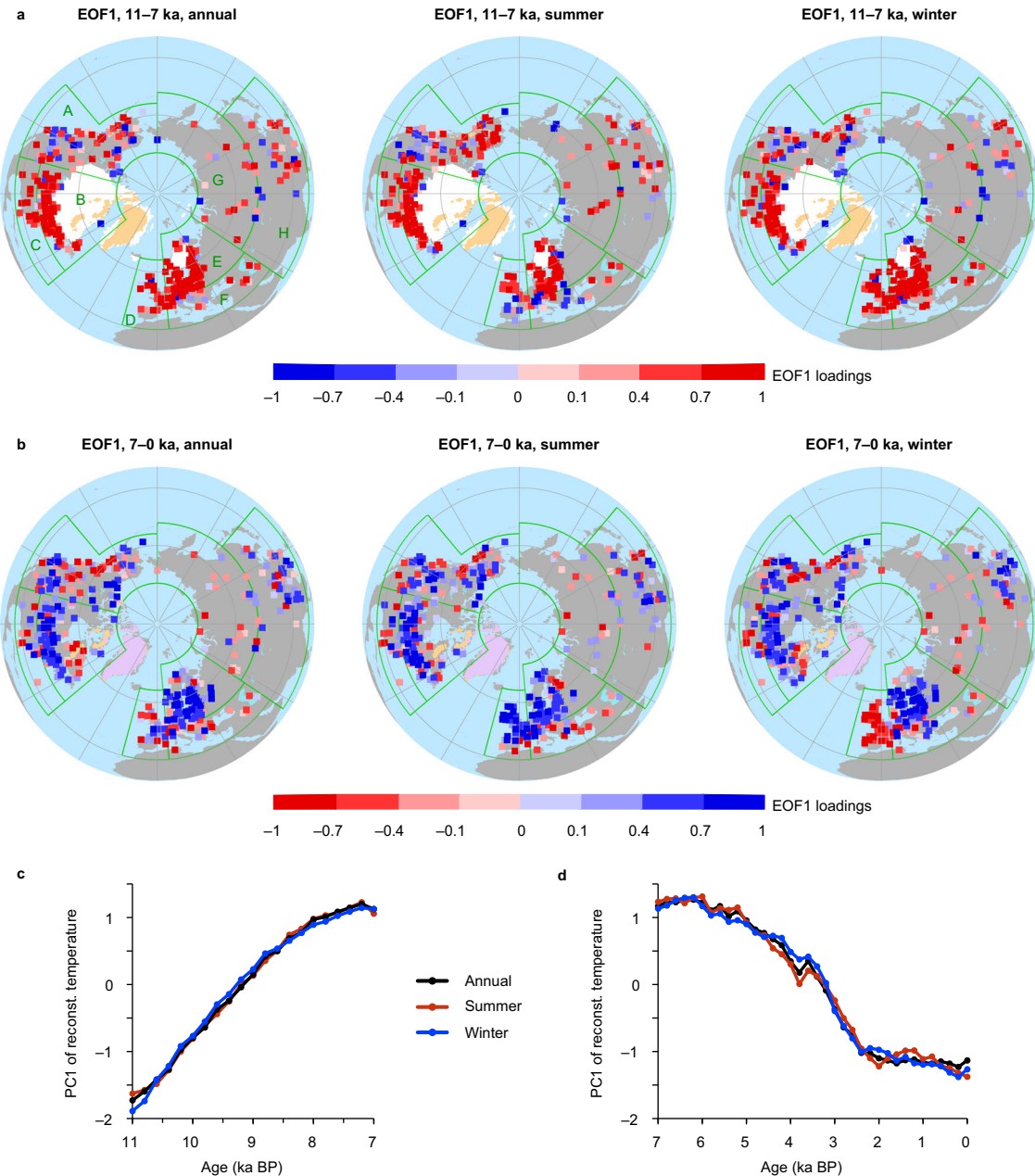

**Fig. 2 | Spatial patterns of Holocene temperature variations revealed by empirical orthogonal function (EOF) analysis. a** EOF1 patterns of annual, summer and winter temperatures from 11 to 7 ka BP. **b** As (**a**) but for period from 7 ka BP to present. The maps show the correlation coefficient between the reconstructed series and PC1 (**c** and **d**) instead of the actual EOF1 values. The percentages of the total variance explained by PC1 are 58%, 52% and 58% for annual, summer and winter temperatures, respectively, during 11–7 ka BP, and 36%, 31% and 42%, respectively, during 7–0 ka BP. The white, yellow and pink shading indicates the ICE-7G ice-sheet range at 11 ka BP, 7 ka BP and at the present, respectively[93]. The green polygons indicate the regions manually divided according to EOF1 patterns and geography: A, western North America (W_NA, 25–75° N & 105–180° W); B, northeastern North America (NE_NA, 40–75° N & 50–105° W); C, southeastern North America (SE_NA, 25–40° N & 50–105° W); D, western Europe (W_EU, 30–65° N & 15°W–5° E); E, northeastern Europe (NE_EU, 48–75° N & 5–56° E); F, southeastern Europe (SE_EU, 30–48° N & 5–56° E); G, northern Asia (N_AS, 50–75° N & 56–180° E); and H, southern Asia (S_AS, 0–50° N & 56–140° E).

For winter and annual temperatures, the warming from ~11 to 7 ka BP is present in both the reconstructions and model results with similar magnitudes (~2.4 and 2.5 °C in annual, and ~4.1 and 3.7 °C in winter for reconstructed and simulated temperatures respectively; Fig. 3a, c), although clear differences exist in western North America and southern Asia where the models show a coherent warming trend, whereas the reconstructions show an HTM phase (Supplementary Figs. 4 and 8). Pronounced discrepancies, however, occur after ~7 ka BP, when the simulated temperature continues to rise until the present (~0.3 and 1.2 °C in annual and winter temperatures, respectively), unlike the pervasive cooling trends in the reconstructions (Fig. 3a, c

and Supplementary Fig. 9). This mismatch during the mid- to late Holocene is also evident in the spatial comparison between the results from Marsicek et al.[9] and the CCSM3 model (Supplementary Figs. 6 and 9), even though they show similar trends in the stack from North America and Europe (see Fig. 2d in Marsicek et al.[9]). Therefore, along with annual temperatures, a conundrum similar to that derived mainly from marine proxies[5] also exists for the winter season at both the hemispheric and regional scales over the NH landmass. The widespread model–data contrast in the spatial comparison also suggests that the uneven distribution of records may be not the main origin of the Holocene conundrum[8].

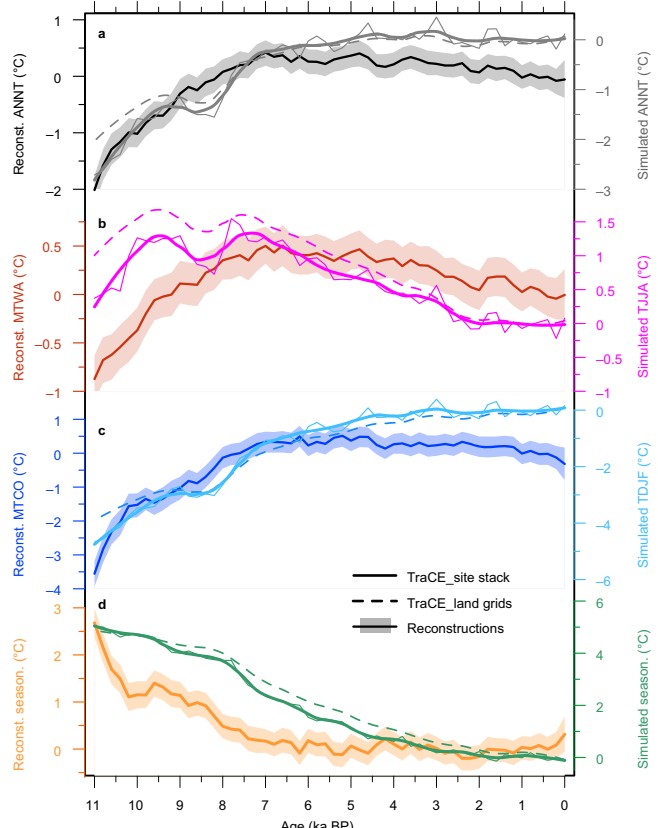

**Fig. 3 | Model–data comparison of Holocene temperature evolution over the Northern Hemisphere (NH) landmass.** Pollen-based and CCSM3-simulated changes of (**a**) annual, (**b**) summer, and (**c**) winter temperatures, and of (**d**) temperature seasonality (summer and winter temperature contrast). Shading indicates 95% uncertainty bands of reconstructions. Simulation results extracted from the same geographical locations as the pollen records are shown by solid lines (thin) with locally-weighted regression smoothing (LOESS; solid thick lines). We also show the LOESS-smoothed simulation results from all NH land grids (dashed lines). The simulation results at pollen sites and all land grids show similar trends, indicating our records can well represent the NH landmass.

With regard to seasonal cycles, however, both the models and reconstructions show a decreasing trend in seasonality, although the amplitude in the simulation is larger as a result of the divergent trends of winter temperature after the mid-Holocene (Fig. 3d).

**Drivers of Holocene temperature change**

Our reconstructions show that Holocene temperature evolution over the NH landmass in both summer and winter seasons was characterised by a warming trend during ~11–7 ka BP, followed by a cooling trend after ~7 ka BP. The early to mid-Holocene warming occurred mainly in eastern North America and Europe; this was successfully simulated by the models (Supplementary Fig. 8). Single forcing experiments from TraCE-21ka indicate that this warming was associated with the retreat of the NH ice sheets (Fig. 4), which can strongly influence temperatures via direct depression and by its effect on surface albedo, meltwater flux, and atmospheric circulation[5,34].

Besides the ice-sheet forcing, single forcing experiments show that orbital-induced seasonal insolation changes play a critical role in seasonal temperature evolutions, resulting in summer cooling and winter warming trends, and thus a long-term decrease in temperature seasonality during the Holocene (Fig. 4b–d). Our reconstructions support the dominant impact of seasonal, especially summer insolation. In regions far from or upwind of the ice sheets, such as southern Asia and western North America, both annual and seasonal

temperatures reached a peak at the early Holocene (Supplementary Fig. 4), which suggests the reduced influence of ice sheets and may reflect the high summer insolation then. After ~7 ka BP, when the Laurentide ice sheet had almost disappeared[35], widespread cooling is evident in the NH during both summer and winter seasons in the reconstructions, which is consistent with the decreasing summer insolation. Therefore, the reconstructed NH seasonal temperature patterns during the Holocene may have been controlled by boreal summer insolation and modulated by NH ice sheets.

Winter insolation can also play an important role in the reconstructed winter temperature changes, which is evident in the TraCE-21ka simulation (Fig. 4c). Single forcing experiments indicate that the reconstructed amplitude of the winter temperature increase from ~11 to 7 ka BP (~4.1 °C) is unlikely to be generated by ice-sheet forcing alone (~1.0 °C), and the rising winter insolation leads to another ~1.0 °C of warming (Fig. 4c). However, the rising winter insolation could not be the dominant factor for the mid-to-late Holocene winter cooling in reconstruction (Fig. 4c). This winter cooling trend may result from the strongly cross-seasonal influence of summer insolation via climatic feedback processes, which we will discuss below. As the contribution of other forcing factors is similar during winter and summer, the reduced insolation seasonality resulted in a long-term decrease in temperature seasonality during the Holocene, which is consistently shown in the reconstructions and simulations (Fig. 4d).

However, the model–data discrepancies in the winter and annual temperature trends after ~7 ka BP are pronounced (Fig. 3a, c and Supplementary Fig. 9), inferring the model–data divergence in temperature response to seasonal insolation and other forcing changes. The mismatch is possibly caused by uncertainties in the reconstructions and/or by model biases related to climate sensitivity and feedbacks[5].

We first examine uncertainties in the pollen-based reconstructions. In addition to temperature, precipitation[21,22,36] and several non-climatic factors[36,37] (e.g., soil, topography, fire and land use) can influence plant growth. These factors may generate "noise" within the overall regional temperature trends (Fig. 2). Nevertheless, our reconstructed NH temperature trends are unlikely biased by a precipitation effect, because the widespread cooling trends after ~7 ka BP (Fig. 2b) differ from the long-term Holocene increase in precipitation in the mid-latitude NH[24] and North America[38].

Another potential reason for the reconstructed mid-to-late Holocene winter and annual cooling trends is that they may represent the warm-season temperatures, as was suggested for the marine-dominant multiproxy reconstructions[5,6]. However, our reconstructions show a good performance in extracting seasonal signals. In regions such as western North America, and western and southern Europe, divergent trends between seasonal temperatures were well reconstructed (Supplementary Figs. 4–6). This performance is also supported by our reconstructed long-term decrease in temperature seasonality (Fig. 3d). In addition, our reconstructions could be more robust for winter than for summer temperatures, because of its more contribution to vegetation variation according to Redundancy Analysis, and the better performance of the calibration for winter temperature indicated by cross-validations (see details in Methods, Supplementary Tables 1 and 2, and Supplementary Data 3).

Alternatively, potential biases in the models may result from imprecise estimation of the climate sensitivity to forcing factors and the incomplete representation of feedback processes[5,39]. Previous studies have suggested that feedbacks in response to insolation forcing, such as vegetation[18,40–43], sea ice[44–47], clouds[18], and dust[48,49], may have large impacts on temperatures, especially winter temperatures[18,41–43,46]. Although most of these feedbacks were incorporated into the TraCE-21ka simulation[5], there are large uncertainties on feedbacks in the models[18,44]. For example, in mid-Holocene

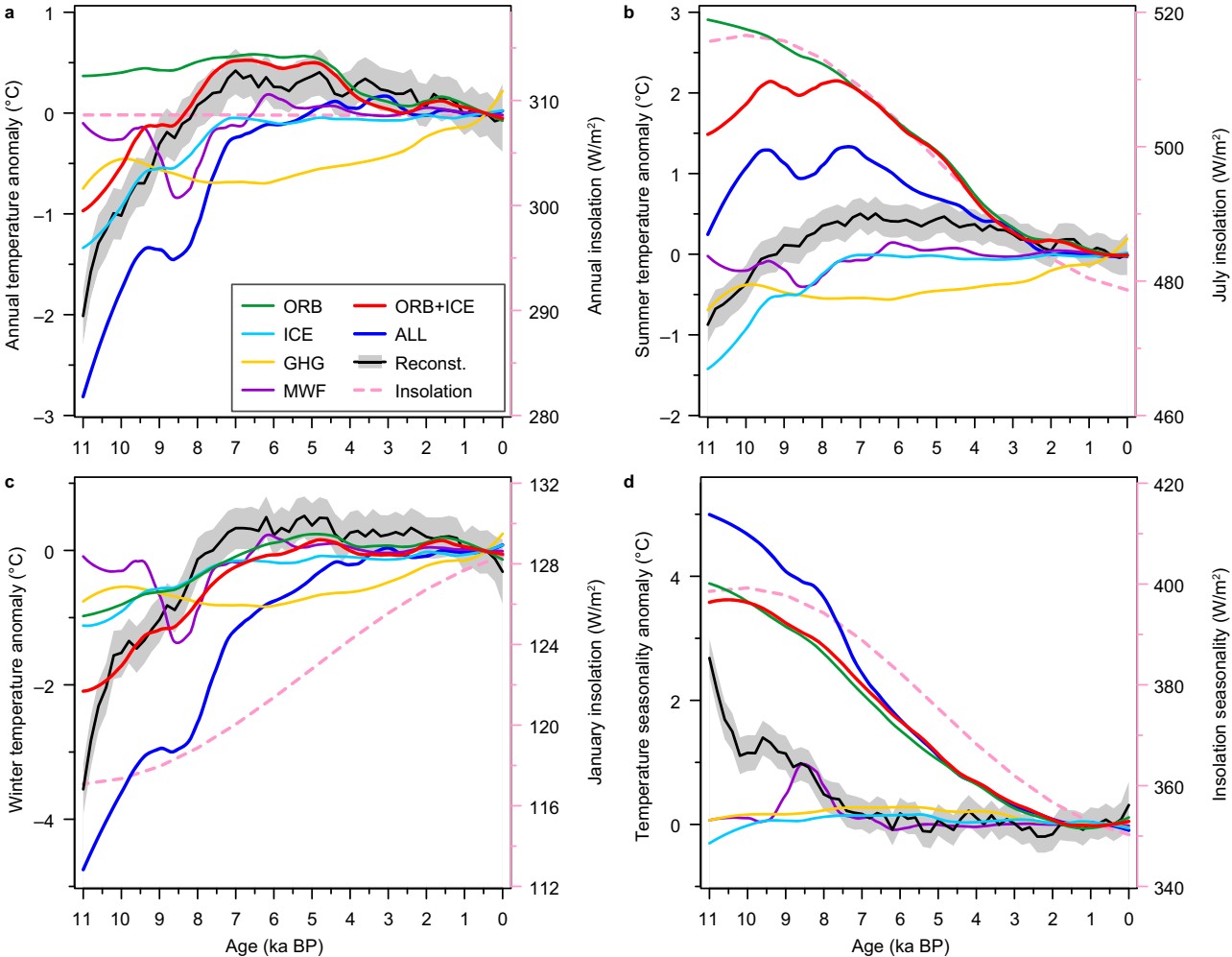

**Fig. 4 | Annual and seasonal temperature trends over the Northern Hemisphere landmass under full and individual forcings from the TraCE-21ka simulation.** Changes of (**a**) annual, (**b**) summer and (**c**) winter temperatures, and of (**d**) temperature seasonality during the Holocene. Forcings include orbital variations (ORB, green), ice sheets (ICE, sky blue), atmospheric greenhouse gases (GHG, yellow), and meltwater fluxes (MWF, purple), as well as the linear sum of ORB and ICE (ORB + ICE, red), and the full forcing simulation (ALL, blue). All simulation results were extracted over the same locations as the pollen sites. The reconstructed temperatures (black line with grey shading showing 95% uncertainty bands) and insolation at 45° N[96] (pink dashed lines) are shown for comparison.

simulations, the vegetation incorporated into the models differs markedly from the "real" vegetation inferred from pollen records, which could contribute -0.3 °C (-0.6 °C) of the warming of annual (winter) temperatures in China[18]. Simulations using pollen-based vegetation in Europe indicate an entirely warmer than pre-industry mid-Holocene during all seasons with the strongest warming in winter[43]. The green Sahara and reduced atmospheric dust loading due to a stronger summer-insolation-induced Afro-Asian monsoon during the mid-Holocene, could increase the global annual mean surface temperature by -0.4 °C[42] and -0.3 °C[48], respectively, which could counteract the -0.3 °C of greenhouse gas-driven decrease relative to the pre-industry[42].

Sea ice feedback is another important process that can influence Holocene temperature evolution[44–47]. The models with stronger sea ice responses to summer insolation changes tend to generate a year-round warmer mid-Holocene in the NH[45], and the sea ice loss-induced winter warming can almost counteract the cooling induced by the lower winter insolation in the mid-high latitudes[46]. Models also indicated an asymmetric pattern of winter temperature responses to Arctic sea ice loss associated with the weakening of mid-high latitude westerlies in the mid-Holocene: the northern Asia cooled, while the Arctic region and North America warmed strongly[46]. Our reconstructions support the importance of sea ice feedbacks by showing a

consistent spatial pattern that there is a warming trend in northern Asia and overall cooling trend in North America after -7 ka BP (Fig. 2 and Supplementary Fig. 4), when the Arctic sea ice coverage increased under the forcing of decreasing summer insolation[50,51].

The impact of feedbacks can be seen in the TraCE-21ka simulation, as the insolation-induced winter (annual) temperatures remain stable (cooling) from -6 ka BP, evidently diverging from the rising winter (unchanged annual) insolation (Fig. 4a, c). As a result, the model–data divergences are reduced when the model considers only orbital and ice-sheet forcing, or if atmospheric greenhouse gas forcing is excluded (Fig. 4a, c). These findings indicate that the feedbacks may be too weak in models to produce the reconstructed annual and winter cooling trends after the mid-Holocene, and/or uncertainty may exist in the climate response to atmospheric greenhouse gas forcing, which was suggested to be the dominant forcing of the mid-late Holocene temperature evolution[5–10].

Bova et al.[6] proposed a solution to the Holocene temperature conundrum based on the assumption that temperatures linearly respond to seasonal insolation changes for oceans between 40° S and 40° N. However, over the NH land, the feedbacks in response to the summer insolation forcing may strongly influence the annual and winter temperature variations and result in the decoupled trends between winter temperature and insolation as the reconstructions

indicate. We suggest that, together with the direct insolation forcing, feedbacks could also be important for the Holocene, as well as for future temperature changes[52–54], and these feedbacks need to be better addressed in future climate modelling studies.

## Methods

### Pollen datasets

The fossil pollen data were obtained from the Neotoma Paleoecology Database[55] (http://www.neotomadb.org/), European Pollen Database[56] (http://www.europeanpollendatabase.net), and the pollen dataset for East Asia[57,58]. A total of 1,310 sites (542 in Europe, 191 in Asia, and 577 in North America; Supplementary Fig. 1 and Supplementary Data 1) were selected using the following criteria to ensure data quality and an adequate site distribution, and thus to reveal reliable long-term temperature trends. (1) The record had more than three chronological controls with at least two independent dates (e.g., radiocarbon). (2) Following the methods of Webb[59], only samples with a date within 1 kyr or bracketed by dates within 6 kyr were retained. (3) The duration of the record exceeded 5 kyr, and the record included samples younger than 1 ka BP, as required to calculate the anomalies. (4) Sampling resolution was finer than 400 years (relaxing to 1000 years for sites in northern Asia to increase the low coverage there). (5) Samples with <200 pollen count were excluded.

To reduce age uncertainty, all radiocarbon dates were recalibrated using the IntCal20 calibration curve[60] and interpolated to sample levels using the R package *clam*[61]. In addition to ages derived from direct dating, other chronological controls, such as annual lamination counts, biostratigraphic controls, and core-top ages from pollen databases[62], were also included in the interpolation.

The modern pollen dataset used in our analysis comprises 13,077 surface samples, compiled from the North America Modern Pollen Database[63], European Modern Pollen Database[64], East Asian Pollen Database[65,66], and the Eurasia pollen database[67] (Supplementary Fig. 1). Modern climate variables, including the mean annual temperature (ANNT) and the temperature of the warmest (MTWA) and coldest months (MTCO), were determined by thin-plate spline interpolation with latitude, longitude and elevation covariates using a 0.5° gridded monthly mean climate dataset (1950–2010, CRU TS v4.01)[68].

### Plant Functional Type (PFT) score calculation

Pollen taxa are commonly used in the reconstruction procedure[9]. However, the taxa-based procedure can be affected by several issues, including lack of modern analogues of fossil assemblages[69], inability to deal with taxa rarely found today, and the influence of non-climatic factors[12]. As an alternative, PFTs (i.e., groups of dominant plants characterised by common phenological and climate constraints[21]) can be used to reduce poor analogue cases and non-climatic influences, and thus provide a more accurate and consistent reflection of the climate[12,69]. Therefore, we used PFT scores in our reconstructions instead of the direct pollen taxa assemblages used in Marsicek et al.[9]. PFT scores were calculated using the taxa–PFT matrixes in biomization procedures for six regions, namely: Europe[70], Eurasia[71], East Asia[72], Canada and the eastern United States[73], the western United States[74] and Beringia[75] (Supplementary Data 2).

### Redundancy analysis (RDA)

Past changes in both summer and winter temperatures have been widely reconstructed using pollen data because of their direct effects on plant growth and distribution[9,12]. However, challenges may arise in the reconstruction of multiple climate variables, which might be biased if they show strong covariance in the calibration data, but this covariance structure may vary over time[76–78]. Summer temperature is frequently regarded as the dominant variable that drives changes in plant composition, mainly according to the studies in the high latitudes[77,78]. However, when considering vegetation biogeography,

cold tolerance and chilling requirements are also critical limiting factors with respect to vegetation dynamics and distribution, and are perhaps more important than summer temperatures[22,23]. To date, it is still insufficient to examine the relative contributions of summer and winter temperatures to vegetation changes.

RDA, a widely used constrained gradient analysis technique, is an effective way to tackle this issue[79,80]. In this study, we used RDA to explore the contributions of seasonal temperatures to the variations in modern PFT score data before reconstructions. Several critical parameters were calculated, including the relative explanatory power (i.e., $\lambda 1/\lambda 2$, the ratio of the constrained to the first unconstrained eigenvalues); the total, unique, and shared proportions of variance explained by each variable; and the significance ($p$ value) of each variable in variation explaining (Supplementary Table 1). RDA was run separately in the six regions defined above for PFT score calculation, using the R package *vegan*[81] (version 2.5–7) and following the procedure presented in Borcard et al.[79].

Our RDA results demonstrate that both MTCO and MTWA are significant variables with respect to the variance of the modern PFT score data in all regions ($p < 0.001$; Supplementary Table 1). Together, they explain more than 20% (23.7% to 26.7%) of the total variance in North America, 18.0% to 19.2% in East Asia and Europe, and 12.1% in Eurasia. The proportions of explanation are fairly high in consideration of the vegetation complexity in large datasets[77]. For each variable, MTCO can explain 9.9% to 20.4% of the total variance, with 5.8% to 13.3% being unique, whereas MTWA explains 6.3% to 20.5%, with 2.3% to 10.2% being unique. The substantial unique contributions of MTCO and MTWA suggest that they have a distinct influence on vegetation changes.

The correlation between MTCO and MTWA (Pearson coefficient $R = 0.35$–$0.76$ in the study regions; Supplementary Table 1) may lead to large covariance and thus induce biases in the reconstructions of the secondary variable[76–78]. Here, we checked the covariance shared by both variables in the partial RDA (Supplementary Table 1). The covariance is indeed positively related to the correlation of the variables. However, in the regions except Canada and the eastern United States where MTCO and MTWA are strongly correlated ($R = 0.76$), the covariance proportions are reasonably low, accounting for 0.9% to 35.8% of the total constrained variance, and are even zero in Beringia, although a moderate correlation ($R = 0.35$) is still present there.

In addition, MTCO generally has stronger explanatory power, explains more of the variance (including total and unique components), and has lower proportions of covariance than MTWA in almost all regions (Supplementary Table 1). This suggests that the winter, rather than the summer temperature is the dominant climatic variable driving vegetation changes in the study regions, which is consistent with the notion of vegetation biogeography[22], and thus it will be more robust for reconstructions of MTCO than MTWA when using PFT scores.

### Modern analogue technique (MAT)

We used the MAT to quantitively reconstruct the seasonal and annual mean temperatures (MTWA, MTCO and ANNT) because of its advantages for large scale reconstructions[9,36,82] and potential power in extracting seasonal temperature signals from PFT score data. Traditional transfer functions (such as weighted averaging-partial least squares, WA-PLS) are commonly used to reconstruct the dominant environmental variable[83]. However, they are susceptible to the covariance problem because they use a constant function calibrated from the modern dataset, but the covariance structure of the variables may change through time[76]. Instead of using constant calibration functions, MAT does not fit pollen–climate response models, but rather, the climate estimates are based on the modern analogues of fossil samples[36,82], and thus can potentially circumvent the covariance problem to some extent. The modern analogues were chosen as the

closest surface sample matches measured using the squared chord distance (SCD)[82], and the dissimilarity-weighted climate of the 5–7 closest modern analogues was then assigned to the fossil sample (Supplementary Table 2).

The underlying assumption of MAT is that an individual vegetation composition (here indicated by PFT scores) is the comprehensive result of a specific combination of causal climatic variables (a climate suite)[82,84]. As each variable has a distinct influence on vegetation, as indicated by the partial RDA results, changes in any variable can result in specific changes to the vegetation composition, and consequently, any given vegetation composition reflects a certain group of variables[84]. Surface pollen records provide a pool of joint vegetation and climate suites. If the fossil records can find close modern analogues, the reconstructions of seasonal temperatures should be robust[12,36].

To ensure that the fossil samples had close modern analogues, three main aspects were considered in this study. Firstly, a spatially dense and extensive modern pollen dataset (Supplementary Fig. 1) was used to provide a large ecological and climatic pool. Secondly, PFT scores, instead of pollen taxa, were used in the dissimilarity calculation[12,69]. Lastly, only fossil samples with close analogues under certain dissimilarity thresholds (Supplementary Table 2) were retained. The thresholds were determined from the trade-off between the precision/accuracy of the reconstruction and the utility for the majority of samples[85]. A relatively low SCD (<0.2 throughout the Holocene) between fossil samples and modern analogues (Fig. 1a) indicates good analogue situations[86], which ensures the reliability of the reconstructions.

The MAT calculation was conducted separately in the six regions defined for the PFTs using the R package *Rioja*[87]. The MAT model performed quite well in the seasonal and annual temperature reconstructions, with a high coefficient of determination ($R^2$, generally > 0.8) based on leave-one-out cross-validation (Supplementary Table 2). We also performed $h$-block cross-validation using the R package *palaeoSig*[88] to assess the influence of spatial autocorrelation[89,90]. In each iteration of validation, the predicting sample and nearby samples within a radius $h$ were omitted. The determination of $h$ is a trade-off between reducing the influence of spatial autocorrelation and the analogue quality[77,78]. Here we proposed a method to find the suitable $h$. Firstly, we conducted a series of cross-validation experiments with $h$ increasing from 0 to 1000 km[78]. Then we matched the median SCD for the Holocene samples with the median SCD curves in the changing-$h$ experiments. The $h$ value with the equivalent SCD was considered to be suitable (Supplementary Fig. 2). In general, the performance of MAT is still good under $h$-block cross-validation (generally, $R^2$ > 0.6 for ANNT and MTCO, and $R^2$ > 0.5 for MTWA), although with lower $R^2$ and higher root-mean-square error of prediction (RMSEP) values than leave-one-out cross-validation (Supplementary Table 2). In addition, MTCO had an overall better performance than MTWA with a higher $R^2$, no matter which cross-validation method was used. This suggests that reconstructions of MTCO could be more robust than MTWA, which agrees with our RDA results.

### Significance testing and isostatic correction of the reconstructions

Statistical significance in RDA and good performance of MAT model in cross-validations reveal that seasonal temperatures can potentially be reconstructed using PFT score data, but it does not guarantee that each reconstructed variable for a specific site is reliable[91]. A significance test for the reconstructions per se was proposed that a significant reconstruction should explain more variance of fossil data than most reconstructions (95%) using random environmental data[91]. The reconstructions failing the test may be questionable and should be treated with caution[91]. However, this test seems to be conservative, because the potentially reliable reconstructions with low variability may be falsely denied (type II errors)[9,36,77,91,92]. Following Marsicek

et al.[9], we adopted a compromised threshold value (one standard deviation, 84.2%) for the significant test using the R package *palaeoSig*[88]. That is, we only retain the reconstructions that explain more variance than 84.2% of the 999 randomly generated reconstructions. After this selection, 1039 of the 1310 sites were retained for at least one variable, including 792 for ANNT, 780 for MTCO, and 740 for MTWA (Supplementary Fig. 1 and Supplementary Data 1). This selection enlarges the amplitude of the stacked NH temperature changes from the early to mid-Holocene (see stacking method below), but has little influence on the temperature trends throughout the Holocene (Supplementary Fig. 3).

The gradual decay of last glacial ice sheets over northern Europe and North America had led to the isostatic adjustment of continental surface as the surface load was removed[93], which may also have important effect on the interpretation of reconstructed temperatures during the Holocene[14]. Following Mauri et al.[14], we calibrated the reconstructed temperatures through subtracting the topography-induced temperature changes. In order to calculate this component, we first used the output of topographic anomalies from the latest ICE-7G_NA (VM7) model[93] (1 degree spatial and 500-year temporal resolution) to obtain the history of topographic changes for each site by spatial and temporal interpretations using inverse distance weighted and linear methods, respectively. We then calculated the topography-induced temperature changes using the present-day locally (within 300 km) topographic lapse rate of temperatures, which were calculated using the CRU climate dataset[68]. After the isostatic correction, the composite temperatures induced by climate change became a bit lower, but with negligible amplitude of variation (Supplementary Fig. 3).

### Temporal interpolation, stacking, and uncertainty analysis

Following Marsicek et al.[9], all of the significant reconstructions after isostatic correction were converted to anomalies using the mean of the latest 1000 years, linearly interpolated to 200-year bins, averaged onto a 2° × 2° grid, and finally generated the hemispheric and regional composites of Holocene seasonal temperatures. The errors produced in courses of reconstruction, temporal interpolation, and stacking were fully considered to estimate the uncertainties in the composites. The sample-specific errors in reconstructions were estimated via bootstrapping the training-set samples for each fossil sample using 100 iterations, and then combining the bootstrap-derived error with the model RMSEP in the leave-one-out cross-validation[87]. In temporal interpolation for each site, we used a Monte Carlo method to randomly sample a normal distribution of the sample-specific errors, and generated an ensemble of 1000 time series to interpolate at 200-year bins separately, from which the median and standard error were derived. Similarly, a 1000-iteration Monte Carlo sampling of the interpolating series with errors was implemented to calculate the median and 95% uncertainty intervals of the final composites.

### Influence of methodological choice on temperature reconstructions

To further evaluate our MAT-based Holocene temperature trends, we conducted reconstructions using two other common methods based on PFT score data, namely a classical transfer function−WA-PLS[83] and a newly-developed machine-learning approach−boosted regression tree (BRT)[77,78]. WA-PLS and BRT were implemented with the R package *Rioja*[78,87] and *gbm*[78,94], respectively. We also generated reconstructions using the MAT method based on pollen taxa data. 60 and 40 taxa were selected for North America and Europe, respectively, following Marsicek et al.[9], while 56 and 74 common taxa from the taxa−PFT matrixes[71,72] were used for northern Asia and East Asia regions, respectively (Supplementary Data 2).

The models' performance indicated by leave-one-out and $h$-block cross-validations can be seen in Supplementary Data 3. Overall, MAT

performs the best in leave-one-out cross-validation for all variables and all regions, and performs similar with BRT in *h*-block cross-validation, while WA-PLS does the worst. These results support the relative advantages of MAT for large-scale reconstructions[9,36]. MAT using PFT scores has comparable performance with MAT using taxa data in Europe and Asia continents, while the latter performs better in North America, which may benefit from the more degrees of freedom in model construction. Additionally, all the methods indicate a generally better performance for MTCO than MTWA with a higher $R^2$ (Supplementary Data 3).

Following the same procedure for MAT, we implemented significance testing, isostatic correction, temporal interpolation and stacking for the new reconstructions. For MAT based on taxa data, 816 records are retained for ANNT (62%), 781 for MTCO (60%), and 821 for MTWA (63%) in the significance-test step. For WA-PLS, the count of significant records is 308 for ANNT (23%), 325 for MTCO (25%) and 286 for MTWA (22%). Fewer than 200 records, however, yielded significant reconstructions for BRT (177 for ANNT, 153 for MTCO, and 149 for MTWA), and thus we skipped the significance selection for BRT to ensure the spatial coverage. This extremely low ratio (<15%) of significant reconstructions warns that the *palaeoSig* approach[88] may be too conservative, as discussed above.

The Holocene temperature trends over the NH landmass reconstructed using MAT (based on both PFTs and taxa data), WA-PLS and BRT methods are generally consistent with each other, with an early to mid-Holocene warming and thereafter cooling trend for both annual and seasonal temperatures (Supplementary Fig. 3), further supporting the robustness of our reconstructions. Inevitably, there is some minor difference among methods. The temperatures reconstructed using WA-PLS has a most pronounced peak at ~7 ka BP, while winter temperature using BRT has a warmer early Holocene than the other twos.

### Compilation of independent temperature reconstructions

We also compare our reconstructions with the quantitative temperature records reconstructed from non-pollen and temperature-sensitive proxies over the NH land (Supplementary Figs. 4–6). Archives were obtained from public paleoclimatic data repositories (NOAA and PANGAEA), previous compilations[2,3,95], and individual publications (Supplementary Data 1, see also in Zhang et al.[4]). As for the pollen data, only records with a sample resolution better than 400 years, with a minimum record duration of 5 kyr, and extending to the Common Era, were retained. A total of 135 independent records was compiled, including 26 of annual and 109 of summer temperatures, but with almost no quantitative reconstructions of winter temperatures. In this study, we only retained the chironomid records ($N = 83$) for regional stacking of the summer temperature in consideration of their robust summer signal[32] (Supplementary Fig. 4).

### TraCE-21ka transient simulation

The global climate over the past 21 kyr has been transiently simulated using a fully coupled atmosphere–ocean general circulation model, CCSM3, developed by the National Center for Atmospheric Research[5]. The TraCE-21ka simulation was synchronously forced by four realistic climate forcings (for further details, see http://www.cgd.ucar.edu/ccr/TraCE): orbital insolation, atmospheric greenhouse gas concentration, continental ice sheets, and meltwater flux. The TraCE-21ka simulation was generally consistent with other models in terms of the Holocene temperature trend[5,44], and the output is available at https://www.earthsystemgrid.org/dataset/ucar.cgd.ccsm3.trace.html. The annual and seasonal temperatures under full and individual forcings were extracted over pollen sites from the output for detailed model–data comparison and analysis of climate drivers.

## Data availability

The pollen data are available from the Neotoma Paleoecology Database (https://www.neotomadb.org), European Pollen Database (https://www.europeanpollendatabase.net) and ref. 57 (https://www.nature.com/articles/s41467-019-09866-8). TraCE-21ka simulations are available at https://www.earthsystemgrid.org/dataset/ucar.cgd.ccsm3.trace.html. Source data are provided with this paper.

## Code availability

All relevant R-packages that were used in this paper are referred to in the "Methods" section. Custom codes used to analyze the data are available from the corresponding author on request.

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

## Acknowledgements

We gratefully acknowledge Zhengyu Liu for his valuable suggestions. Much of the pollen data were obtained from the Neotoma Paleoecology Database (http://www.neotomadb.org) and the European Pollen Database (EPD; http://www.europeanpollendatabase.net/). The work of data contributors, data stewards, and the Neotoma and EPD community is gratefully acknowledged. The work was funded by the National Key Research and Development Program of China (Grant No. 2020YFA0607700), the Strategic Priority Research Program of the Chinese Academy of Sciences (Grant No. XDB26000000), the National Natural Science Foundation of China (Grant Nos. 41888101, 42177180, 41807424, 41572165 and 41690114), and the National Key Research and Development Program of China (Grant No. 2016YFA0600504).

## Author contributions

H.W. and W.Z. conceived the research and designed the study with J.C., H.L. and Z.G. W.Z., J.G. and Q.L. collected the data. W.Z. performed the reconstruction. W.Z., H.W. and Y.Y. analysed the reconstruction results. J.C. and Y.S. analysed the model output. W.Z. wrote the initial manuscript. W.Z., H.W., H.L., J.C., Z.G., Y.S., Y.Y., Q.L. and J.G. commented and contributed to the final version.

## Competing interests

The authors declare no competing interests.

## Additional information

[1]Institute of Geology and Geophysics, Chinese Academy of Sciences, Beijing 100029, China. [2]School of Earth Sciences and Resources, China University of Geosciences (Beijing), Beijing 100083, China. [3]CAS Center for Excellence in Life and Paleoenvironment, Beijing 100044, China. [4]College of Earth and Planetary Sciences, University of Chinese Academy of Sciences, Beijing 100049, China. [5]School of Marine Sciences, Nanjing University of Information Science and Technology, Nanjing 210044, China. [6]School of Geography, Liaoning Normal University, Dalian 116029, China. [7]Key laboratory of Continental Collision and Plateau Uplift, Institute of Tibetan Plateau Research, Chinese Academy of Sciences, Beijing 100101, China. [8]School of Geography and Ocean Science, Nanjing University, Nanjing 210023, China. ✉e-mail: haibin-wu@mail.iggcas.ac.cn; chengjun@nuist.edu.cn

