## [Peer Review File · Nature Communications]

Holocene seasonal temperature evolution and spatial variability over the Northern Hemisphere landmassReviewers' Comments:

Reviewer #1:

Remarks to the Author:

I am excited to see this Holocene temperature reconstruction because previous efforts from the northern hemisphere continents have lacked representation of Holocene changes in Asia, which are obviously important. The paper is an important update on Holocene temperature trends in that regard, and for applying the interesting redundancy analysis. The emphasis on geographic patterns allowed by the EOF analyses is quite illuminating and an important additional contribution.

I like the insight that analyses focused on Europe and eastern North America were strongly influenced by the ice sheets in the early Holocene, but that areas further afield may have had histories more directly tied to the summer insolation trends and that including these other regions highlights different hemispheric-scale temperature trends. (Of course, none of the reconstructions represent temperatures over the ice sheets themselves, which would amplify the cool conditions in the early Holocene and offset the average results shown here). I can well believe that factors mentioned in the main text, such as the representation of Holocene vegetation trends or other feedbacks, may contribute to data-model mismatches.

However, the nuances are lost as currently summarized in the first paragraph (abstract); it sounds as though geographic trends are widely uniform and tied to summer insolation, but not well represented by the models or some reconstructions, which is not the real finding here. Likewise, the final conclusions about feedbacks seems to stretch beyond the analyses here, which are mainly about different regional trends and how they interact to shape the hemispheric mean. The lead paragraph (plus figures, see below) could be reframed to emphasize the core geographic findings. Important regional differences exist both in terms of temperature trends, but also in terms of data-model agreement.

It is not that the models fail completely, they appear to capture the European and North American patterns fairly well. It is only once more geographic coverage is included that the hemispheric divergence develops. If so, it may be useful to explore this difference more thoroughly. Why is the data-model divergence important for Asia? In this way, the results here could be refined to underscore the important message of the recent paper by Osman et al. in Nature that both the Marcott and Temp12k (KA20) reconstructions are affected by the spatial distributions of their samples and do not provide an unbiased mean global temperature reconstruction. I am also not sure that the results here undercut the inference that some datasets have seasonal biases: local comparisons among proxies (e.g., the M18 results cited here – and largely reproduced here at the regional scale as shown in the Extended Data Figure 2) raise questions about the sources of the temperature trends in Atlantic alkenone records; and comparisons between models and proxies (e.g., Bova et al., 2020) appear to affirm the importance of seasonal biases even if the outcome of those biases is similar to the hemisphere-scale outcome here (i.e., the same answer may be developed for multiple reasons).

In this context, I would raise three main points about the manuscript as the authors consider revisions:

First, one of the key messages of the manuscript as represented in the first paragraph (abstract) is that it “reverses a previous pollen-based long-term warming in annual and winter temperatures from North America and Europe” (lines 29-30). This statement is oversimplified and ends up misrepresenting the main findings as described in the main text and Extended Data. Importantly, the pollen-based reconstructions produced here also find winter warming through ~2 ka in southeastern North America, western Europe, and northern Asia (Extended Data Figure 2C). The previous results do not seem “reversed” only contextualized by additional data from other regions.

The main text does a nice job of emphasizing the geographic differences at work in the reconstructed

trends. However, as currently presented, more could be done to show that the different time series in Figure 2 represent different regional outcomes. Since the new results are not substantially different from M18 for Europe and North America (except southeast Europe where M18 replicates cooling during the early Holocene found by multiple previous European analyses, such as Davis and Brewer, 2009 *Climate Dynamics*; Mauri et al., 2015 *QSR*), it would be useful to plot the relevant subset from just Europe and North America with the M18 data in Figure 2a & c. Doing so would show the consistency among studies and highlight that the previously discussed patterns (and model agreement) applies well in those regions. Ideally then, Figure 2 could also complement the European/North American reconstructions with a new set of curves for Asia. In that way, Fig. 2 could highlight global and hemispheric reconstructions at the top and then continental-scale subsets below as a way for highlighting the geographic variability.

The new "IND" reconstruction from selected other proxies either deserves more attention or should just be moved to the Extended Data. As it stands, it mixes annual and summer temperatures, including stable isotope measurements, which are not cleanly a temperature record. If the "IND" reconstruction and its discussion in the text were refined, it would be useful as a synthesis of existing chironomid and biomarker temperatures from the continents, although it is inherently sparse in its geographic coverage.

In this context, it may be better to re-frame the first paragraph here and the conclusions overall to indicate that the inclusion of Asian (and more western North American & Arctic European) pollen data is important for reconstructing a warm early Holocene.

Second, a potentially related point is that M18 produced a time series similar to the one shown here with a peak in the reconstructed temperatures of ~ 0.5 deg C at ca. 7 ka when they simply averaged all of the available reconstructions rather than filtering on significance and gridding the data to avoid overrepresentation of data-dense regions, see their Extended Data Figure 5. The difference is interesting given that this manuscript takes a similar gridding approach, but this study does not filter on significance and therefore ends up with a different spatial distribution of reconstructions, e.g., in western North America. Both this study and M18 applied the palaeoSig package to evaluate the reconstructions, but M18 took a subsequent step, not applied here, of filtering reconstructions based on significance. It is not clear why insignificant reconstructions should be retained. Consider, for example, that tree-ring analyses of the Common Era do not include all tree-ring chronologies, only those with significant relationships to the climate variables of interest. One could argue that the palaeoSig approach has problems for assessing significance (and I would most likely agree that it is too conservative), and therefore, additional sites should be included, but I would be interested in the authors evaluation beyond just asserting that it does not need to be applied. The outcome could be important because (related to the point above) it substantially changes the geographic distribution of sites, including regions like Arctic Europe and western North America. As the authors note, and M18 shown in their Extended Data Figure 5, the site selection has consequences for the reconstructed large-scale temperature trends.

Finally, beyond geography, the methodological decisions used for the pollen-based reconstruction could be usefully evaluated as a source of the differences in this and the previous M18 pollen-based reconstruction. I appreciate the logic behind the PFT based analog reconstruction method but using that approach does substantially reduce the statistical power of the pollen data by collapsing the large number of total pollen taxa to a smaller number of PFTs. It may well be that such an approach is warranted at the hemispheric scale, especially given no-analog pollen assemblages in the early Holocene and later human influences on vegetation (although modern leave-one-out analyses such as shown here apply to a human influenced world and suggest that it may not be a significant concern for the past). However, many degrees of freedom in the multi-variate dataset are lost and some of the calibration/validation statistics are lower than when using the raw pollen taxa. (The methods are not very clear about the number of PFTs used the dissimilarity calculations). I would be curious to know how much of the difference between the new approach here and previous results (such as various

studies finding a cool early Holocene in southern Europe) exists because of the methodological difference, which also includes differences in site and reconstruction selection in Europe and North America.

One added detail is whether the calibration statistics support reconstructing the mean temperature of warmest month versus growing-degree days as done by M18. The two variables are spatially correlated in many regions today, which raises a challenge for using traditional validation approaches to determine what variable is truly represented by the reconstructions. However, the redundancy analyses here may be fruitful for resolving this issue. Notably, the 'biomization' literature cited here for assigning PFTs also argues for an emphasis on bioclimatic variables such as GDDs rather than traditional climate variables. This point seems like a detail except that the CCSM3 TRACE simulations reveal that the mean temperature of the warmest month and GDD5 followed different Holocene trends with different timings of peak warmth – even though they are related summer temperature variables produced by the same model. If proxies such as chironomids track one aspect of summer warmth and pollen another, it would create apparent mismatches with the model depending on the variable reconstructed.

A few details:

One important detail is that the use of many acronyms such as EMH and MLH (plus KA20, M18, etc) is cumbersome and confusing.

Lines 86-87: Osman et al. (2021) should be incorporated here as well. Their data assimilation result confirms the importance of geographic sample locations, but also does not reproduce cooling since 7 ka.

Lines 92-93 emphasize a cooling trend since 7 ka and point out that it is independently validated for summer, but the text then highlights a 'contrast' with evidence of winter warming until 2ka in Europe and North America (line 93-94) as though that summer validation were relevant to winter and as though the winter warming trend on those continents was not found in this study as well.

Why use different baseline periods for the anomalies in different regions (i.e., last 500 yrs for Europe and North America, but 1000 yrs for Asia)? Why not just use 1000 years everywhere?

"All errors were considered" (line 378) but how was the Monte Carlo boot strap method employed to combine the errors? How many iterations?

The manuscript emphasizes that winter temperatures may be the most robust variable reconstructed (line 322-327), but that boreal summer insolation anomalies drove key trends (line 34). I do not understand how summer insolation shaped winter temperatures. This detail in the wording needs better justification or clarification.

Reviewer #2:

Remarks to the Author:

In this paper, Zhang and colleagues present new pollen reconstructions for the Northern Hemisphere extratropical regions. They reconstruct both annual and seasonal changes, and compare these results with the TRACE21K simulations. They present numerous findings, including: 1) the importance of winter temperature on pollen data (and by extension of vegetation), 2) the role of seasonal temperature changes, 3) the drivers of these changes and 4) the discuss potential reasons why models and data disagree.

The paper is interesting and the study very well documented. However, I find it a bit "flat" – pardon the expression. I think the paper could be made more and discussing the consequences of the results

a bit more. There are a lot of interesting results and discussions in the paper about winter temperatures, but these do not get the same attention in the abstract, which focuses more on summer. This is disappointing (but easily fixable!).

I do not have any major criticism about the paper and the analyses that would prevent the publication of this paper. However, the following comments reflect needed/suggested clarifications that should nevertheless be addressed. I do not think they would change the main conclusions of this paper.

Introduction

L 51: ref 11 refers to Scandinavia alone. It is not representative of Europe. Ref 12 is one like. I think this sentence should be rephrased to better illustrate what to say: the HTM has been found across many regions of Europe, Asia and North America.

L59: I would argue that large spatial coverage is necessary but not sufficient. Homogeneous distribution is also important to avoid spatial biases.

L61-63: While I generally agree with this statement, I think it is important to nuance it. Temperature is important in locations where seasonal/annual temperatures impact plant life cycles. In places, water availability can be much more constraining than temperature. In such cases, reconstructing temperatures might be impossible. Please adjust the sentence and specify where this assertion might be valid, e.g. temperate zones?

Methods [I read the methods before reading the results and discussion to avoid biases]

Data selection in such a large-scale study is always tricky and finding the right trade-off between quality and quantity is most complex. While I generally agree with the procedure described here, I can't help it but wanting to see the records from northeastern Russia (fig. SM1) included. This part of the world is so under sampled that even low quality data might provide useful information, wouldn't it?

L274-276: I am not sure I understand this argument. Winter temperature can be reconstructed from the PFTs not because temp is included in the PFTs but because temp impact plants directly. The inclusion of temp in the definition of PFTs is simply a reflection of that, and it doesn't give extra strength to PFTs to reconstruct temperature. Or perhaps I am misunderstanding the argument here? In any case, it needs to be revisited.

L283-291: I fully agree with this paragraph, and it is something that is not sufficiently said in publications. However, I am not sure that the terminology employed here with primary and secondary variables helps. Winter and summer temperature are, in this case, two "primary variables" and as the authors say, it is still complex to determine if one is stronger than the other. I would therefore just stay away from this implicit variable ranking to avoid confusion.

L320: Why should covariance lead to 'underestimation' of the range, and not overestimation?

L334-339: I think the authors use the wrong terminology here. It seems that they are referring to spatial autocorrelation, but they wrote about covariance, which echoes the previous paragraph on the covariance of the studied climate variables. Please adjust.

The distance of 50km is rather small for the h-block analysis. Have the authors tried different thresholds? Also, I am not convinced that a LOO analysis gives really useful results in this type of studies. There are so many samples available to MAT that removing one is not going to have a big effect. It is not wrong, and it can stay in the paper, but I do not think that it is really informative. The HB is much more valuable.

L377: Why not use the same reference for all the records. This is confusing.

L378-381: I cannot understand what the authors did here. It needs some rephrasing and a bit more details.

Did the author correct for the isostatic rebound of northern Europe and Canada? Across the Holocene, the changes in elevation will have a significant effect on the reconstructed temperatures without being driven by climate change per se. See Mauri et al. (2015) [listed in the bibliography already] for one solution to the problem. There are probably other ways. Independently of the method used, accounting for this effect is quite important.

Seasonal Temperature Evolution

The colour scale of fig. 2 should be improved. At the moment, it looks like the vast majority of the points are coloured within either < -0.5 or $> +0.5$. Expanding this a bit to gain better insight of the spatial patterns is required.

Model-data comparison

I think it would be important to note the scaling differences between the data and the models. Changes in TRACE are 2 to 3 times larger than the changes reconstructed from pollen. Might that be a bias in the data, or in the stacking approach?

Drivers of Holocene temperature change

L208-211: I do not understand this argument. Could you explain in greater details why reconstructing different trends in North America and Europe is a validation of an absence of warm-season bias? The trends might well be truly different for the two regions and the proxies still biased towards summer temp (hence a higher capacity to reconstruct small changes and eventually reconstructing divergent trends between regions). Please refine your argument here.

Reviewer #3:

Remarks to the Author:

Reviewer report

NCOMMS-21-43673-T

Holocene seasonal temperature evolution over the Northern Hemisphere landmass

Key results

Accurate records of timing, magnitude, spatial and seasonal aspects of Holocene temperature change are valuable for understanding the relative role of different external and internal climate forcings and their Earth system feedbacks. Zhang et al. provide an extensive stacked reconstruction of Holocene northern hemisphere temperature trends, with focus on the role of seasonality, based on fossil pollen data and methodologically carefully selected and well-validated approaches. They further compare the results to TraCE-21ka model simulation. The results demonstrate, supporting previous reconstructions, that early Holocene warming in both summer and winter temperatures was followed by a thermal maximum around 7 ka and cooling until the pre-industrial era. Prominent and decreasing seasonality (difference between summer and winter temperatures) in the early Holocene was settled by ca. 7 ka. The results, based on the PFT (plant function type) approach, provide an important contribution to the topical discussion of global and large-scale temperature development over the Holocene. Importantly, proxy-based evidence of winter temperature changes support previous proxy-based consensus of late Holocene cooling (see however, recent work by Osman et al. 2021) challenging the capability of present climate models (that show late Holocene warming) in simulating Earth system feedbacks.

Validity

Zhang et al. based their findings on geographically and numerically extensive combination of pollen databases. They carefully evaluate and filter the original data (radiocarbon dates, pollen counts, resolution), their own approaches and results (variation partitioning and RDA, comparison with "individual proxies" and model results over data sites; reasoning for choosing MAT for reconstruction method; Monte Carlo and bootstrap resampling for error estimation).

Significance

There has been a recent increase in global and large-scale Holocene climate reconstructions based on the recent buildup, management and use of extensive open data banks. These are grounded on a mixture or set of proxy sources, some including both or only marine/terrestrial evidence, some on transfer functions (microfossil assemblage based) and geochemical methods (leaf waxes, alkenones, GDGTs), and some on a set of proxy type. The present contribution by Zhang et al. is a very welcome addition, contributing to the very topical discussion on Holocene climate trends and relevant forcings and system feedbacks. Zhang et al. work is important in particular because:

- It is based on an impressive dataset, with a more extensive distribution of sites compared to previous sets (Marcott et al. 2013; Kaufman et al. 2020), in particular from East Asia
- It provides a sound continental-only view, based on one climatically very sound proxy, i.e. pollen, which is then treated carefully through biomization and plant functional types PFTs (there are marine only stacks, but not such large-scale continental only stacks)
- It provides support on proxy (pollen) ability to reconstruct seasonality, challenging model understanding on Earth system feedbacks. This provides direction for future work with great relevance to present day climate science.

Data and methodology

The data quality and validity of used approaches, including MAT, RDA, biomization and stacking, are very high and reasonably well explained and presented in Extended datasets. The selection of PFT over pollen taxa for reconstruction is well described, however see my comments elsewhere on the depth of mechanistic reasoning and discussion.

Analytical approach

The analytical approach from database filtering, use of biomization of pollen data, buildup of state-of-the-art age-depth models (huge effort!), testing of significance of variables and variation partitioning (RDA), as well as testing of Modern Analogue (MAT) based reconstructions (basic performance statistics, leave-one-out, h-block) and error evaluation in the reconstruction stacking procedure. However, I did not find the results of most tests in the files submitted. If this is indeed the case, please provide them in the Supplementaries, regarding MAT testing e.g. closest analogue locations/statistics.

Based on the general knowledge and testing, MAT is probably a good choice for the present large-scale dataset; however, it would have been interesting to see whether other methods such as machine learning or Bayesian based, would have given similar results. I understand such effort might not be feasible for the present manuscript, and as stated above, I consider the approach and evaluation carefully considered. However, I would like to see some discussion on the possible influence of the choice of reconstruction method on the results.

Clarity and context

The manuscript is very easy to read, logically presented and broad audience friendly. There is

sufficient context and appropriate consideration of previous work throughout the text. However, the discussion is simple to the point that I find it superficial in places, which gives a (false?) impression that the authors have not considered the context and interpretations fully. The authors use appropriate references, but the reader is not going to go searching for explanations in the original publications; the authors are supposed to state their own assessment and mechanistic interpretations. A few examples:

Lines 38-40 While it is a good idea to start with a general sentence, there could be a bit more depth to this; e.g. with respect to importance of forcings and feedbacks of the system.

Lines 140-142 This is a very interesting result and put into context of previous work; however the reasons are not discussed from climate forcing/feedback perspective here or in the later section on drivers.

Lines 186-190 The authors provide no mechanistic evidence/explanation why they think the NH seasonal temperature patterns and trends would have been controlled by boreal summer insolation, modulated by ice sheets.

Lines 202-203 What are these non-climatic factors?

Line 208 What do you mean by warm season bias, how it is expressed and why it might be a "missing" bias

Lines 213-214 Why is winter temperature more critical than summer temperature or precipitation (over NH landmass)? Is this "critical influence" true across the NH and all vegetation types?

Suggested improvements

Key suggestions:

- See comments below on the lack of depth and mechanistic approach in discussion that should be improved throughout the manuscript.
- Please include Osman et al. 2021 Nature in the discussion and reference list

Other minor suggestions, see also comments in other sections of review report:

Lines 64-66 Explain here or elsewhere, how your choice of method (MAT) might affect the results, as compared to if you chose WA-PLS, machine learning or Bayesian methods. There is work that combines different methods, see e.g. [Salonen, J.S., Korpela, M., Williams, J.W. et al. Machine-learning based reconstructions of primary and secondary climate variables from North American and European fossil pollen data. *Sci Rep* 9, 15805 (2019). <https://doi.org/10.1038/s41598-019-52293-4>]

Lines 252-253 <200 pollen count (the number of pollen counted, not the number of grains present)
Line 268 This is really not true; while some taxa such as aquatic species can be excluded, most taxa-based quantitative reconstructions used in published regional and global compilation reconstructions have used most available (not few indicator) taxa. Please revise.

Lines 305-306 Is around 20% explanation of the total variance a lot? How about the rest of the 80% of variance that influence the reconstructions? Please elaborate a bit on this for example by providing context from other studies or methodology.

Lines 673-64 I would not amplify the reconstruction seasonality, why would you do this? If you do so, then please provide secondary axis. As is, the figure will be incorrectly interpreted.

Figure 1. Please provide ref. MA18 summer temperature reconstruction for comparison and discuss it. This is a GDD >5 °C reconstruction; not sure why this is not presented and could not find explanation in text.

Response to Reviewers

We really appreciate the constructive comments and suggestions from the reviewers. We believe that these are quite important to improve the quality of our manuscript. Substantial revisions are made according to the comments and point-by-point responses are listed below.

REVIEWER COMMENTS

Reviewer #1 (Remarks to the Author):

I am excited to see this Holocene temperature reconstruction because previous efforts from the northern hemisphere continents have lacked representation of Holocene changes in Asia, which are obviously important. The paper is an important update on Holocene temperature trends in that regard, and for applying the interesting redundancy analysis. The emphasis on geographic patterns allowed by the EOF analyses is quite illuminating and an important additional contribution.

Responses: Thank you for your encouraging comments on our work.

I like the insight that analyses focused on Europe and eastern North America were strongly influenced by the ice sheets in the early Holocene, but that areas further afield may have had histories more directly tied to the summer insolation trends and that including these other regions highlights different hemispheric-scale temperature trends. (Of course, none of the reconstructions represent temperatures over the ice sheets themselves, which would amplify the cool conditions in the early Holocene and offset the average results shown here). I can well believe that factors mentioned in the main text, such as the representation of Holocene vegetation trends or other feedbacks, may contribute to data-model mismatches.

Responses: Thank you for your encouraging comments.

However, the nuances are lost as currently summarized in the first paragraph (abstract); it sounds as
though geographic trends are widely uniform and tied to summer insolation, but not well represented
by the models or some reconstructions, which is not the real finding here. Likewise, the final
conclusions about feedbacks seems to stretch beyond the analyses here, which are mainly about
different regional trends and how they interact to shape the hemispheric mean. The lead paragraph
(plus figures, see below) could be reframed to emphasize the core geographic findings. Important
regional differences exist both in terms of temperature trends, but also in terms of data-model
agreement.

**Responses:** Thank you for your enlightening comments. We agree with you that the regional
divergence is an important part of our findings. It can lead to divergent trends among the temperature
reconstructions which use records with different geographic coverage. Our extensive dataset
provides a way to reconcile the discrepancies to some extent. We have reorganized the abstract and
emphasized the spatial heterogeneity of temperature changes. Please see details in line 23-35 in the
revised manuscript, also as follows:

*“Our results indicate that both summer and winter temperatures warmed from the early to mid-
Holocene (~11–7 ka BP) and then cooled thereafter, but with significant spatial variability. Strong
early Holocene warming trend occurred mainly in Europe, eastern North America and northern
Asia, which can be generally captured by model simulations and is likely associated with the retreat
of continental ice sheets. The subsequent cooling trend is pervasively recorded except for northern
Asia and southeastern North America regions, which may reflect the cross-seasonal impact of the
decreasing summer insolation through climatic feedbacks, but the cooling in winter season is not
well reproduced by climate models.”*

However, another important finding in our study is that widespread cooling trends after the mid-
Holocene happened in both summer and winter seasons over the Northern Hemisphere (NH)
landmass except for regions, such as northern Asia and southeastern North America (see detailed
discussions in the next item). The cooling trend in winter season may result from several feedback
processes (such as vegetation, sea ice, etc.) in response to the summer insolation forcing (see detailed
discussions in line 597-631 in this document). However, the widespread winter cooling trends are not
reproduced by climate model simulations, which may reflect the uncertainties in feedback processes
in models. Please review our discussions on this issue in the revised manuscript (line 261-295).

It is not that the models fail completely, they appear to capture the European and North American
patterns fairly well. It is only once more geographic coverage is included that the hemispheric
divergence develops. If so, it may be useful to explore this difference more thoroughly. Why is the
data-model divergence important for Asia? In this way, the results here could be refined to
underscore the important message of the recent paper by Osman et al. in Nature that that both the
Marcott and Temp12k (KA20) reconstructions are affected by the spatial distributions of their
samples and do not provide an unbiased mean global temperature reconstruction. I am also not sure
that the results here undercut the inference that some datasets have seasonal biases: local
comparisons among proxies (e.g., the M18 results cited here – and largely reproduced here at the
regional scale as shown in the Extended Data Figure 2) raise questions about the sources of the
temperature trends in Atlantic alkenone records; and comparisons between models and proxies (e.g.,
Bova et al., 2020) appear to affirm the importance of seasonal biases even if the outcome of those
biases is similar to the hemisphere-scale outcome here (i.e., the same answer may be developed for
multiple reasons).

**Responses:** We gratefully appreciate for your detailed and enlightening suggestions. Marsicek et al.
(2018) ever indicated that models could well capture the pollen-based seasonal temperatures at
Europe and North America continents via comparisons of the stack curves. However, they did not
compare them in the spatial aspect.

We carefully checked the results of simulation and our reconstruction in Europe and North America.
The model-data divergences in annual and winter temperatures after the mid-Holocene are widely
distributed over the Northern Hemisphere land, including Europe and North America regions (see
details in the revised Supplementary Fig. 9 and a copy Fig. R1 below in this document).

**The spatial divergence between simulations and Marsicek et al. (2018) is also evident, especially**
**in Europe and northeastern North America** (see details in the revised Supplementary Fig. 6 and 9,
and a copy Fig. R2 below in this document), which can also be seen in the stack curves (Fig. R3
below in this document). Although the discrepancies among model and data in Asia region (mainly
southern Asia) is evident, the widespread discrepancies in North America and Europe regions are
also obvious (Figs. R1-R3).

The paper by Osman et al. (2021) is a great work that they produced a proxy-constrained, full-field
reanalysis of global surface temperature changes. Our results are in line with the conclusions of
Osman et al. (2021) that spatial distributions of records have important influence on the
reconstructions, and it is important to have a more geographic coverage. In the study of Osman et al.

(2021), they used 539 proxy records mainly at the marginal ocean to constrain the multi-model
outputs. However, we concern that the records may be not enough, considering the absence of
records from the vast land areas. Our extensive dataset over the Northern Hemisphere landmass is
useful for a more unbiased mean global temperature reconstruction using the data assimilation
method of Osman et al. (2021).

With regards to the seasonal bias issue, our reconstructions show that the mid- to late Holocene
cooling trends happened in both summer and winter, as well as annual temperatures over the
Northern Hemisphere land (revised Fig. 1), suggesting that the temperatures may response
nonlinearly to the seasonal insolation change, via feedback processes, such as vegetation and sea ice
(see details in line 261-286 in the revised manuscript). The nonlinear response breaks the primary
assumptions of Bova et al. (2021) that temperatures linearly respond to the seasonal insolation
changes. Therefore, our results don't support their conclusions that the model-data mismatch is
mainly originated from seasonal bias in proxies, at least over the Northern Hemisphere land.

Certainly, we could not infer whether the marine proxy records, such as Atlantic alkenone records,
have problems about seasonal biases based on our study, but our results suggest that the potential
biases of feedback processes in models may be a source of model-data mismatch.

In addition, previous studies focusing on the seasonality issue indicate that alkenone proxy can well
represent the annual mean sea surface temperatures in the majority of global oceans, including the
Atlantic oceans (e.g., Rosell-Melé and Prah, 2013; Tierney and Tingley, 2018). Summer biases
indeed occur in the high-latitude North Atlantic (Tierney and Tingley, 2018), where there are only
two alkenone records in the study of Marcott et al. (2013), and meanwhile, the multi-model
simulations (PMIP4-CMIP6 and PMIP3-CMIP5) show a year-round warmer-than-pre-industry mid-
Holocene there (Brierley et al., 2020). The majority of North Atlantic alkenone records in Marcott et
al. (2013) locates at the mid-latitudes (9 records), especially Mediterranean region (6 records).
However, in Mediterranean region, the alkenone proxy may bias towards winter season, rather than
summer season (Tierney and Tingley, 2018). Therefore, these lines of evidence from the proxy
reexamination also don't support the conclusions proposed by Bova et al. (2021) that warm-season
biases in proxies are mainly responsible for the model-data discrepancies.

Fig. R1: Model–data comparisons of annual and winter temperature patterns for 7–0 ka BP (a copy from the revised Supplementary Fig. 9).

Fig. R2: Spatial comparisons between reconstructions of Marsicek et al. (2018) and TraCE-21ka simulation (a copy from the revised Supplementary Fig. 6 and 9).

Fig. R3: Comparisons between reconstructions and TraCE-21ka simulation in Europe and northeastern North
 America (NE_NA).

References:

Marsicek, J., Shuman, B. N., Bartlein, P. J., Shafer, S. L. & Brewer, S. Reconciling divergent trends
 and millennial variations in Holocene temperatures. *Nature* 554, 92–96 (2018).

Osman, M. B. et al. Globally resolved surface temperatures since the Last Glacial Maximum. *Nature*
 599, 239–244 (2021).

Bova, S., Rosenthal, Y., Liu, Z., Godad, S. P. & Yan, M. Seasonal origin of the thermal maxima at
 the Holocene and the last interglacial. *Nature* 589, 548–553 (2021).

Rosell-Melé, A. & Prah, F. G. Seasonality of UK'37 temperature estimates as inferred from
 sediment trap data. *Quat. Sci. Rev.* 72, 128–136 (2013).

Tierney, J. E. & Tingley, M. P. BAYSPLINE: A New Calibration for the Alkenone
 Paleothermometer. *Paleoceanography and Paleoclimatology* 33, 281–301 (2018).

Marcott, S. A., Shakun, J. D., Clark, P. U. & Mix, A. C. A reconstruction of regional and global
 temperature for the past 11,300 years. *Science* 339, 1198–1201 (2013).

Brierley, C. M. et al. Large-scale features and evaluation of the PMIP4-CMIP6 midHolocene
 simulations. *Clim. Past* 16, 1847–1872 (2020).

In this context, I would raise three main points about the manuscript as the authors consider
revisions:

First, one of the key messages of the manuscript as represented in the first paragraph (abstract) is that
it “reverses a previous pollen-based long-term warming in annual and winter temperatures from
North America and Europe” (lines 29-30). This statement is oversimplified and ends up
misrepresenting the main findings as described in the main text and Extended Data. Importantly, the
pollen-based reconstructions produced here also find winter warming through ~2 ka in southeastern
North America, western Europe, and northern Asia (Extended Data Figure 2C). The previous results
do not seem “reversed” only contextualized by additional data from other regions.

**Responses:** Thank you for the suggestion. We thoroughly revised this part following your
comments. We deleted this sentence and emphasized the spatial variability of temperature change.
As you pointed out, there are regions with a long-term winter warming in southeastern North
America, western Europe, and northern Asia. However, the Northern Hemisphere land shows an
overall cooling trend after ~7 ka BP (the revised Fig. 1), because the majority of North America,
Europe and southern Asia is characterized by a cooling trend, as is clarified in the above item.

Please see details in line 23-35 in the revised manuscript.

*“Our results indicate that both summer and winter temperatures warmed from the early to mid-
Holocene (~11–7 ka BP) and then cooled thereafter, but with significant spatial variability. Strong
early Holocene warming trend occurred mainly in Europe, eastern North America and northern
Asia, which can be generally captured by model simulations and is likely associated with the retreat
of continental ice sheets. The subsequent cooling trend is pervasively recorded except for northern
Asia and southeastern North America regions, which may reflect the cross-seasonal impact of the
decreasing summer insolation through climatic feedbacks, but the cooling in winter season is not
well reproduced by climate models.”*

The main text does a nice job of emphasizing the geographic differences at work in the reconstructed
trends. However, as currently presented, more could be done to show that the different time series in
Figure 2 represent different regional outcomes. Since the new results are not substantially different
from M18 for Europe and North America (except southeast Europe where M18 replicates cooling
during the early Holocene found by multiple previous European analyses, such as Davis and Brewer,
2009 Climate Dynamics; Mauri et al., 2015 QSR), it would be useful to plot the relevant subset from

just Europe and North America with the M18 data in Figure 2a & c. Doing so would show the
consistency among studies and highlight that the previously discussed patterns (and model
agreement) applies well in those regions. Ideally then, Figure 2 could also complement the
European/North American reconstructions with a new set of curves for Asia. In that way, Fig. 2
could highlight global and hemispheric reconstructions at the top and then continental-scale subsets
below as a way for highlighting the geographic variability.

**Responses:** Thank you. This suggestion is very helpful to clarify the regional temperature patterns
and the differences among reconstructions. We added the temperature stacks of Europe and North
America, and Asia in the revised Fig. 1 (a copy Fig. R4 below in this document). For further details,
we added the individual stacks of Europe and North America in the revised Supplementary Fig. 4 (a
copy Fig. R5 below in this document).

In Europe and North America, our reconstructions indicated that all the annual and seasonal
temperature trends well replicate the trends over the Northern Hemisphere landmass, both of which
show a mid-Holocene peak (Fig. R4). This indicates that Europe and North America matter
substantially in the temperature trends over the Northern Hemisphere landmass.

Our reconstructed annual temperature trend is similar with MA18, although our peaking time (~7 ka
BP) is ~2 kyr earlier (Fig. R5). For winter temperature, however, our result is evidently different
from MA18, which shows a long-term warming until ~2 ka BP (Fig. R5). The divergence between
MA18 and our reconstructions is derived from the difference in North America, which is further
attributed to the divergence in western North America, since MA18 and our reconstructions have
similar trends in Europe and eastern North America (Fig. R5).

In western North America, we have much more records than MA18, and thus have a comprehensive
view on the Northern Hemisphere temperature variations (see site distribution in Supplementary Fig.
1 and 6, and also Fig. R6 below in this document). The nearly undetected annual cooling trend and
the long-term winter warming trend after ~7 ka BP in MA18 may result from their scarcity of records
from western North America and the influence of the strong warming trends in southeastern North
America. Please see these discussions in line 136-151 in the revised manuscript.

*“The divergence of annual temperature trends between our results and Marsicek et al.⁹ in North
America and Europe (Fig. 1a) is mainly derived from the difference in western North America, since
similar trends were found in eastern North America and Europe (Supplementary Figs. 4–6). In
western North America, our much more records can provide a comprehensive view, and indicate an
HTM pattern, rather than a long-term warming trend⁹ (Supplementary Figs. 4–6). Therefore, the*

nearly undetected cooling trend after ~7 ka BP from Marsicek et al.⁹ (Fig. 1a) may result from the
 influence of the strong warming trends in southeastern North America, considering their scarcity of
 records from western North America, together with the absence of data from Asia (Supplementary
 Figs. 4 and 6).

A similar situation is evident for the winter season. EOF1 patterns and regional stacks from eastern
 North America and Europe are generally consistent between our results and Marsicek et al.⁹
 (Supplementary Figs. 4–6), which are also consistent with another pollen-based reconstruction from
 Europe¹⁴ (Supplementary Fig. 4). With the denser distribution of records in western North America
 and the new data from Asia, the winter temperature over the NH landmass developed a cooling trend
 after ~7 ka BP, which contrasts with the long-term warming trend until ~2 ka BP, suggested by
 Marsicek et al.⁹ (Fig. 1c).”

Fig. R4: Multiproxy records of annual and seasonal temperature trends during the Holocene in the NH (a copy
 from the revised Fig. 1).

Fig. R5: Comparisons of annual and winter temperatures among reconstructions in North America and Europe
 (a copy from the revised Fig. 1 and Supplementary Fig. 4). Regions: North America and Europe (NAEU),
 western (W_NA), northeastern (NE_NA), southeastern (SE_NA) North America.

Fig. R6: Distribution of significant sites in North America from Marsicek et al (2018) and this study.

The new “IND” reconstruction from selected other proxies either deserves more attention or should
just be moved to the Extended Data. As it stands, it mixes annual and summer temperatures,
including stable isotope measurements, which are not cleanly a temperature record. If the “IND”
reconstruction and its discussion in the text were refined, it would be useful as a synthesis of existing
chironomid and biomarker temperatures from the continents, although it is inherently sparse in its
geographic coverage.

**Responses:** Thank you for this comment. Following your suggestion, we removed the independent
reconstruction from the revised Fig. 1 (see also in R4 above in this document), and retained only
chironomid records for the regional stacks (see the revised Supplementary Fig. 4). Please see the
corresponding modifications in line 158-169 and 536-538 in the revised manuscript. For the
biomarker records, such as brGDGTs, they are commonly used as proxies for mean annual
temperature, but seasonal bias may happen and some other environmental factors, such as soil pH,
can also strongly influenced these proxies (e.g., De Jonge et al., 2021). Therefore, we don't use the
biomarker records in the regional stacking as well.

*“Although there is an overall agreement in Holocene summer temperature trend^{3,4,9,14} (Fig. 1b),*
*some divergence exists between the reconstructions using chironomids and pollen data during the*
*early Holocene. That is, the former tends to show a warmer early Holocene than the latter in*
*northeastern Europe, western North America and northern Asia regions (Supplementary Fig. 4). The*
*divergence could be partly attributed to spatial inconsistencies, since the EOF1 patterns are overall*
*consistent, but the chironomid records have much lower site density in the mid-latitudes and are*
*dominated by the cooling trend at the high latitudes (Supplementary Fig. 5). Proxy uncertainties may*
*also be an important source of the discrepancy^{28–30}. The pollen-based early Holocene temperature*
*may be underestimated due to the delayed tree growth after the long-distance migration to newly*
*glacial retrieved areas²⁸, while chironomids may be influenced by some other factors than*
*temperature, such as water depth and nutrient availability^{29,30}.”*

*“In this study, we only retained the chironomid records (N = 83) for regional stacking of the summer*
*temperature in consideration of their robust summer signal³² (Supplementary Fig. 4).”*

**References:**

De Jonge, C. et al. The influence of soil chemistry on branched tetraether lipids in mid- and high
latitude soils: Implications for brGDGT- based paleothermometry. *Geochimica et*
*Cosmochimica Acta* 310, 95–112 (2021).

In this context, it may be better to re-frame the first paragraph here and the conclusions overall to
indicate that the inclusion of Asian (and more western North American & Arctic European) pollen
data is important for reconstructing a warm early Holocene.

**Responses:** Thank you. We have reframed the abstract paragraph and emphasized the spatial
variability of temperature variations. Please see details in line 21-35 in the revised manuscript.
However, as was discussed above, the warm mid-Holocene was widespread over Northern
Hemisphere land, except for northern Asia and southeastern North America regions (Figs. R1 and
R4).

*“Here we present an extensive dataset of Holocene seasonal temperatures reconstructed using 1,310*
*pollen records covering the Northern Hemisphere landmass. Our results indicate that both summer*
*and winter temperatures warmed from the early to mid-Holocene (~11–7 ka BP) and then cooled*
*thereafter, but with significant spatial variability. Strong early Holocene warming trend occurred*
*mainly in Europe, eastern North America and northern Asia, which can be generally captured by*
*model simulations and is likely associated with the retreat of continental ice sheets. The subsequent*
*cooling trend is pervasively recorded except for northern Asia and southeastern North America*
*regions, which may reflect the cross-seasonal impact of the decreasing summer insolation through*
*climatic feedbacks, but the cooling in winter season is not well reproduced by climate models.”*

Second, a potentially related point is that M18 produced a time series similar to the one shown here
with a peak in the reconstructed temperatures of ~0.5 deg C at ca. 7 ka when they simply averaged
all of the available reconstructions rather than filtering on significance and gridding the data to avoid
overrepresentation of data-dense regions, see their Extended Data Figure 5. The difference is
interesting given that this manuscript takes a similar gridding approach, but this study does not filter
on significance and therefore ends up with a different spatial distribution of reconstructions, e.g., in
western North America. Both this study and M18 applied the palaeoSig package to evaluate the
reconstructions, but M18 took a subsequent step, not applied here, of filtering reconstructions based
on significance. It is not clear why insignificant reconstructions should be retained. Consider, for
example, that tree-ring analyses of the Common Era do not include all tree-ring chronologies, only
those with significant relationships to the climate variables of interest. One could argue that the
palaeoSig approach has problems for assessing significance (and I would most likely agree that it is
too conservative), and therefore, additional sites should be included, but I would be interested in the
authors evaluation beyond just asserting that it does not need to be applied. The outcome could be

important because (related to the point above) it substantially changes the geographic distribution of
sites, including regions like Arctic Europe and western North America. As the authors note, and M18
shown in their Extended Data Figure 5, the site selection has consequences for the reconstructed
large-scale temperature trends.

**Responses:** Thank you for the suggestion. Significance test of the reconstructions suggested by
Telford and Birks (2011) is useful to judge whether one reconstruction is significantly different from
most of the randomized reconstructions (95%). However, it is susceptible to several type II errors
(false negative), which means that the potentially reliable reconstruction is denied by the test
(Telford and Birks, 2011). Type II errors tend to happen in situations such as (1) sites with relative
low number of species; (2) sites with no adequate samples; and (3) the reconstructions with low
climate variability (Telford and Birks, 2011).

The *palaeoSig* only tests if there is a statistically significant trend in the data relative to an ensemble
of random reconstructions; but it does not test whether the reconstructed values make sense
climatically (Chevalier et al., 2020). Failing the test does not necessarily mean the reconstruction is
incorrect, and the results should be considered alongside other palaeoecological and environmental
evidence (e.g., Brooks et al., 2012; Salonen et al., 2014; Payne et al., 2016; Lemmen et al., 2018;
Chevalier et al., 2020).

Although the *palaeoSig* test may be too conservative, it suggests that the reconstructions failing the
test should be treated more cautiously (e.g., Telford and Birks, 2011; Brooks et al., 2012; Lemmen et
al., 2018). As the reviewer's suggestion, we conduct the *palaeoSig* test and only retain the
reconstructions that explain more variations than 84.2% of the 999 randomly generated
reconstructions, following Marsicek et al. (2018). However, the composite temperature trends show
little difference, although the amplitudes enlarge, especially during the early to mid- Holocene (see
the revised Supplementary Fig. 3 and a copy Fig. R7 below in this document).

In addition, we still have much more records (N=382 for ANNT and 363 for MTCO) than Marsicek
et al. (2018) (N=217 for ANNT and 200 for MTCO) in North America, especially in western North
America, after the significant tests and strict quality control (Fig. R6 above in this document),
because our dataset from NEOTOMA is an updated version (October 2018) of Marsicek et al. (2018)
(July 2013).

Please see details in line 447-462 in the revised manuscript.

*“Statistical significance in RDA and good performance of MAT model in cross-validations reveal*
*that seasonal temperatures can potentially be reconstructed using PFT score data, but it does not*

*guarantee that each reconstructed variable for a specific site is reliable*⁹¹. *A significance test for the*
 *reconstructions per se was proposed that a significant reconstruction should explain more variance*
 *of fossil data than most reconstructions (95%) using random environmental data*⁹¹. *The*
 *reconstructions failing the test may be questionable and should be treated with caution*⁹¹. *However,*
 *this test seems to be conservative, because the potentially reliable reconstructions with low*
 *variability may be falsely denied (type II errors)*^{9,36,77,91,92}. *Following Marsicek et al.*⁹, *we adopted a*
 *compromised threshold value (one standard deviation, 84.2%) for the significant test using the R*
 *package palaeoSig*⁸⁸. *That is, we only retain the reconstructions that explain more variance than*
 *84.2% of the 999 randomly generated reconstructions. After this selection, 1,039 of the 1,310 sites*
 *were retained for at least one variable, including 792 for ANNT, 780 for MTCO, and 740 for MTWA*
 *(Supplementary Fig. 1 and Supplementary Data 1). This selection enlarges the amplitude of the*
 *stacked NH temperature changes from the early to mid-Holocene (see stacking method below), but*
 *has little influence on the temperature trends throughout the Holocene (Supplementary Fig. 3)."*

 **Fig. R7: Influence of significance selection on Holocene temperature trends. Red lines indicate the Northern**
 **Hemisphere stack of significant sites, while black lines indicate the stack of all sites.**

References:

- Telford, R. J. & Birks, H. J. B. A novel method for assessing the statistical significance of
quantitative reconstructions inferred from biotic assemblages. *Quaternary Sci. Rev.* 30, 1272–
1278 (2011).
- Chevalier, M. et al. Pollen-based climate reconstruction techniques for late Quaternary studies.
*Earth-Sci. Rev.* 210, 103384 (2020).
- Salonen, J. S. et al. Reconstructing palaeoclimatic variables from fossil pollen using boosted
regression trees: comparison and synthesis with other quantitative reconstruction methods.
*Quat. Sci. Rev.* 88, 69–81 (2014).
- Brooks, S. J., Axford, Y., Heiri, O., Langdon, P. G. & Larocque-Tobler, I. Chironomids can be
reliable proxies for Holocene temperatures. A comment on Velle et al. (2010). *The Holocene*
22, 1495–1500 (2012).
- Payne, R. J. et al. Significance testing testate amoeba water table reconstructions. *Quaternary Sci.*
*Rev.* 138, 131–135 (2016).
- Lemmen, J. & Lacourse, T. Fossil chironomid assemblages and inferred summer temperatures for the
past 14,000 years from a low-elevation lake in Pacific Canada. *J. Paleolimnol.* 59, 427–442
(2018)
- Marsicek, J., Shuman, B. N., Bartlein, P. J., Shafer, S. L. & Brewer, S. Reconciling divergent trends
and millennial variations in Holocene temperatures. *Nature* 554, 92–96 (2018).

Finally, beyond geography, the methodological decisions used for the pollen-based reconstruction
could be usefully evaluated as a source of the differences in this and the previous M18 pollen-based
reconstruction. I appreciate the logic behind the PFT based analog reconstruction method but using
that approach does substantially reduce the statistical power of the pollen data by collapsing the large
number of total pollen taxa to a smaller number of PFTs. It may well be that such an approach is
warranted at the hemispheric scale, especially given no-analog pollen assemblages in the early
Holocene and later human influences on vegetation (although modern leave-one-out analyses such as
shown here apply to a human influenced world and suggest that it may not be a significant concern
for the past). However, many degrees of freedom in the multi-variate dataset are lost and some of the
calibration/validation statistics are lower than when using the raw pollen taxa. (The methods are not
very clear about the number of PFTs used the dissimilarity calculations). I would be curious to know

how much of the difference between the new approach here and previous results (such as various
studies finding a cool early Holocene in southern Europe) exists because of the methodological
difference, which also includes differences in site and reconstruction selection in Europe and North
America.

**Responses:** The advantages of PFT over taxa data in large-scale reconstructions have been clarified
in line 334-340 in the revised manuscript. PFT can reduce poor analogue cases, non-climatic
influences, and thus provide a more accurate and consistent reflection of the climate (e.g., Davis et
al., 2003; Peyron et al., 1998). In addition, the use of PFT scores increases the number of pollen taxa
that can be used, including those that do not occur in the modern calibration dataset, through
combining them into PFTs (Davis et al., 2003). In fact, when using taxa data, only partially selected
taxa, rather than all the species, were used to reduce the issues of overfitting to the non-climatic
signals (Juggins et al., 2015). For example, Marsicek et al. (2018) used 60 selected taxa for North
America and 40 taxa for Europe. In our study, we used 22 PFTs for Europe (grouped from 79 taxa),
27 for Eurasia (100 taxa), 36 for East Asia (266 taxa), 26 for Canada and the eastern United States
(50 taxa), 31 for western United States (69 taxa), and 25 for Beringia (101 taxa) (see details in
Supplementary Data 2).

We have shown that **the temperature trends are generally similar between our reconstructions**
**using PFTs and Marsicek et al. (2018) using taxa data in eastern North America and the**
**majority of Europe except for southeastern Europe, indicating the methodological decision has**
**limited influence on the temperature trends** (see details in the revised Supplementary Fig. 4).

We also conducted reconstructions using MAT based on pollen taxa data to further clarify the
influence of methodological choice on temperature reconstructions (see details in line 498-501 in the
revised manuscript). In North America and Europe regions, we use the same taxa with Marsicek et
al. (2018). In Asia regions, we used the common taxa in PFT-taxa matrix (see details in the revised
Supplementary Data 2). As expected, the results show that there is little difference between
reconstructions using PFT score and pollen taxa data in the composite Holocene temperature trends
over the Northern Hemisphere land (see the revised Supplementary Fig. 3 and a copy Fig. R8 below
in this document).

In southeastern Europe, we added another pollen-based reconstruction from Mauri et al. (2015) for
comparison, which is the updated version of Davis et al. (2003) and has source data at
<https://www.ncei.noaa.gov/access/paleo-search/study/18317>. We also compared them with our
reconstructions using the taxa data (Fig. R9 below in this document).

The results of Mauri et al. (2015) and Marsicek et al. (2018) show opposite trends of Holocene
 temperatures. The former shows long-term warming trends in all seasons with a cool early Holocene,
 whereas the latter shows a much warmer early Holocene (the revised Supplementary Fig. 4 and a
 copy Fig. R9 below in this document). **Our results using both PFTs and taxa data show a cool**
 **early Holocene in winter season, but a warm-than-present condition in summer season.** We
 have no idea about the exact reason of the difference among these temperature reconstructions in
 southeastern Europe (difference in site distribution may be a reason), but it could not result from the
 methodology whether PFT or taxa data were used, because our results using PFTs and taxa data
 show similar trends (Fig. R9).

Our reconstructions seem to capture the seasonal difference better in this area, and agree better with
 the independent reconstructions, such as Milandre Cave fluid inclusion record of annual mean
 temperature (Affolter et al., 2019) and chironomid-based summer temperature (e.g., Samartin et al.,
 2017) (Fig. R9). Please see details in line 170-174 in the revised manuscript.

*“A local-scale divergence was ever found in southeastern Europe, namely a long-term year-round*
 *warming¹⁴ versus cooling⁹ throughout the Holocene (Supplementary Fig. 4). Our new*
 *reconstructions capture an evident seasonal difference, that is, a warmer-than-present summer but a*
 *cooler-than-present winter during the early Holocene, and agree better with the independent*
 *records^{31–33} (Supplementary Fig. 4).”*

Fig. R8: Holocene temperature variations over the Northern Hemisphere landmass reconstructed using MAT
 based on the PFT score (red) and pollen taxa data (magenta).

 Fig. R9: Comparisons of Holocene temperature variations in southeastern Europe.

**References:**

Davis, B. A. S., Brewer, S., Stevenson, A. C. & Guiot, J. The temperature of Europe during the
 Holocene reconstructed from pollen data. *Quat. Sci. Rev.* 22, 1701–1716 (2003).
 Peyron, O. et al. Climatic reconstruction in Europe for 18,000 yr B.P. from pollen data. *Quat. Res.*
 49, 183–196 (1998).
 Juggins, S., Simpson, G. L. & Telford, R. J. Taxon selection using statistical learning techniques to
 improve transfer function prediction. *The Holocene* 25, 130–136 (2015).
 Mauri, A., Davis, B. A. S., Collins, P. M. & Kaplan, J. O. The climate of Europe during the
 Holocene: a gridded pollen-based reconstruction and its multi-proxy evaluation. *Quat. Sci. Rev.*
 112, 109–127 (2015).

Affolter, S. et al. Central Europe temperature constrained by speleothem fluid inclusion water
isotopes over the past 14,000 years. *Sci. Adv.* 5, v3809 (2019).

Samartin, S. et al. Warm Mediterranean mid-Holocene summers inferred from fossil midge
assemblages. *Nat. Geosci.* 10, 207–212 (2017).

One added detail is whether the calibration statistics support reconstructing the mean temperature of
warmest month versus growing-degree days as done by M18. The two variables are spatially
correlated in many regions today, which raises a challenge for using traditional validation approaches
to determine what variable is truly represented by the reconstructions. However, the redundancy
analyses here may be fruitful for resolving this issue. Notably, the ‘biomization’ literature cited here
for assigning PFTs also argues for an emphasis on bioclimatic variables such as GDDs rather than
traditional climate variables. This point seems like a detail except that the CCSM3 TRACE
simulations reveal that the mean temperature of the warmest month and GDD5 followed different
Holocene trends with different timings of peak warmth – even though they are related summer
temperature variables produced by the same model. If proxies such as chironomids track one aspect
of summer warmth and pollen another, it would create apparent mismatches with the model
depending on the variable reconstructed.

**Responses:** Thank you for this comment. GDD₅ is indeed an important bioclimatic factor for plant
growth. GDD₅ and the summer temperature are commonly correlated to each other. Following your
suggestion, we conducted RDA analysis to clarify the relative contributions of summer temperature
(MTWA) and GDD₅, as well as winter temperature (MTCO), on vegetation variations (Table R1
below in this document).

In general, MTCO has the strongest explanatory power (higher λ_1/λ_2) and explains more of the
variance (including total and unique components). MTWA explains more of the variance than GDD₅
in North America while GDD₅ explains more in Europe and Asia. The Pearson correlation shows
that MTWA and GDD₅ indeed strongly correlate to each other (>0.8 for all regions except East
Asia). However, the covariance of vegetation shared by MTWA and GDD₅ is generally equivalent
to, or even smaller than the one shared by MTCO and GDD₅. As a result, MTWA has generally more
unique contributions of vegetation variation than GDD₅. The results suggest that MTWA is a more
independent variable than GDD₅, when we have chosen MTCO, and the reconstruction of MTWA
may be less influenced by the covariance problem suggested by Juggins et al. (2013). This is

consistent with the fact that GDD₅ is complex in seasonal aspect; that is, it may represent seasons
from spring to autumn, and even winter in the middle and low latitudes.

The mismatch during the early Holocene between reconstructions using pollen and other biotic
proxies (such as chironomids) at northern high latitudes has been discussed in previous studies. It
was sometimes attributed to the delayed tree establishment to climate change due to the long-
distance migration to newly glacial retrieved areas (e.g., Välranta et al., 2015). The chironomids also
have shortcomings that they may respond to some other environmental factors than temperature,
such as water depth, pH and nutrient availability (e.g., Luoto et al., 2014; Shala et al., 2017;
McKeown et al., 2019).

It is also possible that the pollen reconstruction may represent GDD₅, rather than summer
temperature, as the reviewer's suggestion. We have checked the Holocene trends of summer
temperature and GDD₅ from TraCE-21ka simulation over pollen sites in five regions where the
majority of chironomid records locate (namely, western North America, western Europe, northeastern
Europe, southeastern Europe, and northern Asia; Fig. R10 below in this document). The result indicates
that the summer temperature and GDD₅ from TraCE-21ka show similar trends in almost all the five
regions except southeastern Europe, where our reconstruction is more consistent with the simulated
summer temperature with an early Holocene peak than the simulated GDD₅ with a mid-Holocene
peak (Fig. R10).

We added some discussions about the mismatch between pollen and chironomids in line 158-169 in
the revised manuscript.

*“Although there is an overall agreement in Holocene summer temperature trend^{3,4,9,14} (Fig. 1b),*
*some divergence exists between the reconstructions using chironomids and pollen data during the*
*early Holocene. That is, the former tends to show a warmer early Holocene than the latter in*
*northeastern Europe, western North America and northern Asia regions (Supplementary Fig. 4). The*
*divergence could be partly attributed to spatial inconsistencies, since the EOF1 patterns are overall*
*consistent, but the chironomid records have much lower site density in the mid-latitudes and are*
*dominated by the cooling trend at the high latitudes (Supplementary Fig. 5). Proxy uncertainties may*
*also be an important source of the discrepancy²⁸⁻³⁰. The pollen-based early Holocene temperature*
*may be underestimated due to the delayed tree growth after the long-distance migration to newly*
*glacial retrieved areas²⁸, while chironomids may be influenced by some other factors than*
*temperature, such as water depth and nutrient availability^{29,30}.”*

Both of the GDD₅ and MTWA have been widely reconstructed using pollen data, for example, for
 both (e.g., Bartlein et al., 2011), GDD₅ only (e.g., Marsicek et al., 2018), and summer temperature
 only (e.g., Davis et al., 2003; Seppa et al., 2009; Mauri et al., 2015; Salonen et al., 2019). In this
 study, we intend to reveal the Holocene variations of seasonal temperatures and the seasonality, and
 thus we prefer the MTWA.

Table R1: Contributions of MTCO, MTWA and GDD₅ to the variation of modern PFT score data
 based on RDA. *R* is Pearson correlation coefficient. λ_1/λ_2 represents the relative explanatory power
 of each variable. *Explained* and *Unique* are the whole and unique proportions of total variation
 explained by each variable in the partial RDA. *p* is the significance value of each variable.

Region	Variable	R	λ_1/λ_2	Explained (%)	Unique (%)	p
Europe	MTCO		0.92	14.9	5.5	0.001
	MTWA		0.60	10.9	1.3	0.001
	GDD ₅		0.87	13.7	1.3	0.001
	MTCO&MTWA	0.65		19.0	0.2	
	MTCO&GDD ₅	0.74		19.0	2.7	
	MTWA&GDD ₅	0.85		14.9	2.8	
	ALL			20.3	6.9	
Eurasia	MTCO		0.51	9.9	4.0	0.001
	MTWA		0.31	6.3	1.5	0.001
	GDD ₅		0.37	7.4	0.9	0.001
	MTCO&MTWA	0.57		12.1	0.1	
	MTCO&GDD ₅	0.70		11.5	1.9	
	MTWA&GDD ₅	0.87		9.0	0.7	
	ALL			13.0	4.0	
East Asia	MTCO		0.68	13.4	4.3	0.001
	MTWA		0.37	7.9	4.0	0.001
	GDD ₅		0.61	11.8	1.7	0.001
	MTCO&MTWA	0.54		18.0	0.4	
	MTCO&GDD ₅	0.74		15.8	5.8	
	MTWA&GDD ₅	0.62		15.4	0.6	
	ALL			19.7	3.7	
Canada and the eastern United States	MTCO		1.15	20.4	6.1	0.001
	MTWA		1.02	20.5	4.2	0.001
	GDD ₅		0.78	17.1	2.6	0.001
	MTCO&MTWA	0.76		25.9	1.1	
	MTCO&GDD ₅	0.80		24.3	0.8	
	MTWA&GDD ₅	0.88		22.4	1.4	
	ALL			28.5	13.9	
western United States	MTCO		0.82	17.9	10.8	0.001
	MTWA		0.87	16.5	2.7	0.001
	GDD ₅		0.77	15.7	2.8	0.001

	MTCO&MTWA	0.64		26.7	0.3	
	MTCO&GDD ₅	0.68		26.8	0.7	
	MTWA&GDD ₅	0.89		18.7	6.1	
	ALL			29.5	7.5	
Beringia	MTCO		0.57	13.6	5.1	0.001
	MTWA		0.36	10.5	4.3	0.001
	GDD ₅		0.56	14.0	5.0	0.001
	MTCO&MTWA	0.28		23.9	5.5	
	MTCO&GDD ₅	0.59		24.6	8.2	
	MTWA&GDD ₅	0.85		23.7	6.1	
	ALL			28.8	5.2	

Fig. R10: Holocene trends of summer temperature and GDD₅ from TraCE-21ka simulation, along with the
 reconstructions using pollen and chironomids data. W_NA, western North America; W_EU, western Europe;
 NE_EU, northeastern Europe; SE_EU, southeastern Europe; N_AS, northern Asia.

References:

Juggins, S. Quantitative reconstructions in palaeolimnology: new paradigm or sick science? *Quat.*
*Sci. Rev.* 64, 20–32 (2013).

Väliranta, M. et al. Plant macrofossil evidence for an early onset of the Holocene summer thermal
maximum in northernmost Europe. *Nat. Commun.* 6, (2015).

Luoto, T. P., Kaukolehto, M., Weckström, J., Korhola, A. & Väliranta, M. New evidence of warm
early-Holocene summers in subarctic Finland based on an enhanced regional chironomid-based
temperature calibration model. *Quaternary Res.* 81, 50–62 (2014).

Shala, S. et al. Comparison of quantitative Holocene temperature reconstructions using multiple
proxies from a northern boreal lake. *Holocene* 27, 1745–1755 (2017).

McKeown, M. M. et al. Complexities in interpreting chironomid-based temperature reconstructions
over the Holocene from a lake in Western Ireland. *Quaternary Sci. Rev.* 222, 105908 (2019).

A few details:

One important detail is that the use of many acronyms such as EMH and MLH (plus KA20, M18,
etc) is cumbersome and confusing.

**Responses:** Thank you. Following your suggestion, we have revised these acronyms to common
expression throughout the revised manuscript.

Lines 86-87: Osman et al. (2021) should be incorporated here as well. Their data assimilation result
confirms the importance of geographic sample locations, but also does not reproduce cooling since 7
543 ka.

**Responses:** Thank you. We have added this reference. Please see details in line 96-97 in the revised
manuscript.

*“The long-term cooling trend after ~7 ka BP has been widely detected in syntheses of proxy*
*records^{2–4,24–26}, in global/NH ocean temperature records^{2,3,8}, and accords with the general pattern of*
*NH glacier advances²⁷, but it is not well shown in the previous pollen-based reconstruction for North*
*America and Europe⁹ (MA18, Fig. 1a), and in a model–data assimilation output of global surface*
*temperature constrained by marine proxy records⁸.”*

Lines 92-93 emphasize a cooling trend since 7 ka and point out that it is independently validated for
summer, but the text then highlights a ‘contrast’ with evidence of winter warming until 2ka in
Europe and North America (line 93-94) as though that summer validation were relevant to winter and
as though the winter warming trend on those continents was not found in this study as well.

**Responses:** Thank you. We have rephrased this sentence. Please see details in line 101-106 in the
revised manuscript.

*“For the winter temperature, the warming trend from ~11 to 7 ka BP is in accord with*
*reconstructions of Marsicek et al.⁹, but the subsequent cooling trend is opposite to them, which*
*demonstrated a continuous warming until ~2 ka BP in North America and Europe (Fig. 1c).”*

Why use different baseline periods for the anomalies in different regions (i.e., last 500 yrs for Europe
and North America, but 1000 yrs for Asia)? Why not just use 1000 years everywhere?

**Responses:** Thank you. We take your advice and use 1000 years as the baseline period for all
regions. Please see details in line 315-317 and 478-479 in the revised manuscript.

*“The duration of the record exceeded 5 kyr, and the record included samples younger than 1 ka BP,*
*as required to calculate the anomalies.”*

*“Following Marsicek et al.⁹, all of the significant reconstructions after isostatic correction were*
*converted to anomalies using the mean of the latest 1000 years”*

“All errors were considered” (line 378) but how was the Monte Carlo boot strap method employed to
combine the errors? How many iterations?

**Responses:** Thank you. Please see the revisions (line 481-491 in the revised manuscript) as follows:

*“The errors produced in courses of reconstruction, temporal interpolation, and stacking were fully*
*considered to estimate the uncertainties in the composites. The sample-specific errors in*
*reconstructions were estimated via bootstrapping the training-set samples for each fossil sample*
*using 100 iterations, and then combining the bootstrap-derived error with the model RMSEP in the*
*leave-one-out cross-validation⁸⁷. In temporal interpolation for each site, we used a Monte Carlo*
*method to randomly sample a normal distribution of the sample-specific errors, and generated an*
*ensemble of 1000 time series to interpolate at 200-year bins separately, from which the median and*

*standard error were derived. Similarly, a 1000-iteration Monte Carlo sampling of the interpolating*
*series with errors was implemented to calculate the median and 95% uncertainty intervals of the*
*final composites.”*

The manuscript emphasizes that winter temperatures may be the most robust variable reconstructed
(line 322-327), but that boreal summer insolation anomalies drove key trends (line 34). I do not
understand how summer insolation shaped winter temperatures. This detail in the wording needs
better justification or clarification.

**Responses:** Thank you for this comment. This seems indeed a confusing thing because it is more
acceptable that seasonal temperatures may linearly response to the seasonal insolation changes (e.g.,
Bova et al., 2021). The winter insolation can certainly play an important role in the Holocene winter
temperature changes (see in line 226-233 in the revised manuscript). “insolation” may be better than
“boreal summer insolation” here in the abstract. We have deleted “boreal summer” in the sentence.
Please see details in line 36 in the revised manuscript. “...highlight the important impact of
*insolation and associated feedbacks on temperature changes, which warrant closer attention in*
*future climate modelling.”*

As is discussed above, however, the reconstructed winter temperatures show a widespread cooling
trends after the mid-Holocene over the Northern Hemisphere land, which is opposite to the rising
winter insolation, but consistent with the summer insolation. In the reconstruction aspect, we have
shown that our reconstructions could be more robust for the winter temperature, because of its more
contribution to vegetation variation according to Redundancy Analysis, and the better performance
of the calibration for winter than for summer temperatures (see detailed discussions in line 255-260,
389-394, 441-444 and 508-510 in the revised manuscript).

Along with the direct seasonal insolation forcing, previous studies have suggested that several
feedback processes in response to insolation forcing, such as vegetation (e.g., Ganopolski et al.,
1998; Davies et al., 2015; Lin et al., 2019; Tabor et al., 2020), sea ice (e.g., Park et al., 2018, 2019;
Zhang et al., 2021) and dust (Davies et al., 2015; Liu et al., 2018), may have large impacts on
temperatures, especially winter temperatures (Lin et al., 2019; Tabor et al., 2020; Park et al., 2018).

For example, the green Sahara due to a stronger summer-insolation-induced Afro-Asian monsoon
during the mid-Holocene, results in 0.39 °C of global annual mean surface warming, which
counteracts the 0.29 °C of greenhouse gas-driven cooling relative to the pre-industry (Tabor et al.,
2020). The green Sahara-induced warming is most pronounced at winter seasons and at the high

latitudes (Tabor et al., 2020). Additionally, the reduced atmospheric dust loading due to a green
Sahara during the mid-Holocene could also increase the global annual mean surface temperature by
~ 0.3 °C (Liu et al., 2018).

Another sensitive simulation in the mid-Holocene show that when the reconstructed vegetation using
pollen records were incorporated into the models, the annual and winter temperatures could increase
by ~ 0.3 °C and ~ 0.6 °C, respectively, in China (Lin et al., 2019). Similar situations occurred in
Europe. Strandberg et al. (2022) simulated the mid-Holocene Europe climate using the reconstructed
vegetation based on pollen records and using three models (one fully coupled general circulation
model, EC-Earth version 3.1, and two regional climate models, namely Rossby Centre Atmosphere
model and HCLIM38-ALADIN model). All of their simulations show that the reconstructed
vegetation have a significant impact on climate change; the mid-Holocene was warmer than pre-
industry during all seasons for practically all of Europe; and the most pronounced warming happened
in winter season (Strandberg et al., 2022).

Sea ice feedback is another important process that can influence Holocene temperature evolution
(e.g., Park et al., 2018, 2019; Zhang et al., 2021). The enhanced summer solar heating during the
mid-Holocene reduced year-round Arctic sea ice cover, resulting in a significant warming of the mid-
and high latitudes in winter season (Park et al., 2019). The models with stronger sea-ice responses to
summer insolation changes tend to generate a year-round warmer mid-Holocene in the NH,
especially in the Arctic and North America regions (Park et al., 2018).

Please see these discussions in detail in line 261-295 in the revised manuscript.

References:

Bova, S., Rosenthal, Y., Liu, Z., Godad, S. P. & Yan, M. Seasonal origin of the thermal maxima at
the Holocene and the last interglacial. *Nature* 589, 548–553 (2021).

Ganopolski, A., Kubatzki, C., Claussen, M., Brovkin, V. & Petoukhov, V. The influence of
vegetation-atmosphere-ocean interaction on climate during the mid-Holocene. *Science* 280,
1916–1919 (1998).

Lin, Y. et al. Mid-Holocene climate change over China: model–data discrepancy. *Clim. Past* 15,
1223–1249 (2019).

Davies, F. J., Renssen, H., Blaschek, M. & Muschitiello, F. The impact of Sahara desertification on
Arctic cooling during the Holocene. *Clim. Past* 11, 571–586 (2015).

Tabor, C., Otto Bliesner, B. & Liu, Z. Speleothems of South American and Asian Monsoons
Influenced by a Green Sahara. *Geophys. Res. Lett.* 47, (2020).

Strandberg, G. et al. Mid-Holocene European climate revisited: New high-resolution regional climate
model simulations using pollen-based land-cover. *Quat. Sci. Rev.* 281, 107431 (2022).

Park, H. S., Kim, S. J., Stewart, A. L., Son, S. W. & Seo, K. H. Mid-Holocene Northern Hemisphere
warming driven by Arctic amplification. *Sci. Adv.* 5, x8203 (2019).

Park, H. et al. The impact of Arctic sea ice loss on mid-Holocene climate. *Nat. Commun.* 9, 4571
(2018).

Zhang, X. & Chen, F. Non-trivial role of internal climate feedback on interglacial temperature
evolution. *Nature* 600, E1–E3 (2021).

Liu, Y. et al. A possible role of dust in resolving the Holocene temperature conundrum. *Sci. Rep.* 8,
4434 (2018).

Reviewer #2 (Remarks to the Author):

In this paper, Zhang and colleagues present new pollen reconstructions for the Northern Hemisphere
extratropical regions. They reconstruct both annual and seasonal changes, and compare these results
with the TRACE21K simulations. They present numerous findings, including: 1) the importance of
winter temperature on pollen data (and by extension of vegetation), 2) the role of seasonal
temperature changes, 3) the drivers of these changes and 4) the discuss potential reasons why models
and data disagree.

**Responses:** Thank you very much for your encouraging comments.

The paper is interesting and the study very well documented. However, I find it a bit “flat” – pardon
the expression. I think the paper could be made more and discussing the consequences of the results
a bit more. There are a lot of interesting results and discussions in the paper about winter
temperatures, but these do get the same attention in the abstract, which focuses more on summer.
This is disappointing (but easily fixable!).

**Responses:** Thank you. As you pointed out, the reconstruction of winter temperature is critical to
reveal the source of model-data divergence in Holocene temperature trends. We have reframed the
abstract. Please see details in line 21-38 in the revised manuscript.

*“Here we present an extensive dataset of Holocene seasonal temperatures reconstructed using 1,310*
*pollen records covering the Northern Hemisphere landmass. **Our results indicate that both summer***
***and winter temperatures warmed from the early to mid-Holocene (~11–7 ka BP) and then cooled***
***thereafter**, but with significant spatial variability. Strong early Holocene warming trend occurred*
*mainly in Europe, eastern North America and northern Asia, which can be generally captured by*
*model simulations and is likely associated with the retreat of continental ice sheets. The subsequent*
*cooling trend is pervasively recorded except for northern Asia and southeastern North America*
*regions, which may reflect the cross-seasonal impact of the decreasing summer insolation through*
*climatic feedbacks, **but the cooling in winter season is not well reproduced by climate models. Our***
*results challenge the proposal that seasonal biases in proxies are the main origin of model–data*
*discrepancies and highlight the critical impact of insolation and associated feedbacks on*
*temperature changes, which warrant closer attention in future climate modelling.”*

I do not have any major criticism about the paper and the analyses that would prevent the publication
of this paper. However, the following comments reflect needed/suggested clarifications that should
nevertheless be addressed. I do not think they would change the main conclusions of this paper.

**Responses:** Thank you for your valuable comments and suggestions.

Introduction

L 51: ref 11 refers to Scandinavia alone. It is not representative of Europe. Ref 12 is one lqke. I think
this sentence should be rephrased to better illustrate what to say: the HTM has been found across
many regions of Europe, Asia and North America.

**Responses:** Thank you. We have rephrased this sentence. Please see details in line 53 in the revised
manuscript.

*“The HTM in summer season has been found across many regions of Europe^{12–14}, Asia¹⁵, and North*
*America¹⁶, as is shown in multiproxy stacks^{3,4} (KA20, Fig. 1b)”*

L59: I would argue that large spatial coverage is necessary but not sufficient. Homogeneous
distribution is also important to avoid spatial biases.

**Responses:** We agree with you that homogeneous distribution is also important, and have added this
aspect in line 62-64 in the revised manuscript. *“it is important to use reliable records with a large
spatial coverage and **fine geographic representation** to comprehensively understand Holocene
climate change⁸.”*

However, we have to say that homogeneity is inevitably limited by the real site distribution. To
increase the homogeneity as much as possible, we relax the selection criteria for the northern Eurasia
region where pollen sites are relatively sparse (see details in line 317-319 in the revised manuscript,
*“Sampling resolution was finer than 400 years (relaxing to 1000 years for sites in northern Asia to
increase the low coverage there).”*). In addition, we take a 2° × 2°-grid-averaged procedure before
hemispheric and regional stacks to reduce the influence of inhomogeneity (line 480 in the revised
manuscript).

L61-63: While I generally agree with this statement, I think it is important to nuance it. Temperature
is important in locations where seasonal/annual temperatures impact plant life cycles. In places,
water availability can be much more constraining than temperature. In such cases, reconstructing
temperatures might be impossible. Please adjust the sentence and specify where this assertion might
be valid, e.g. temperate zones?

**Responses:** Thank you. Temperatures influence plant growth not only in temperate zones, but also in
boreal and tropical zones. However, as you point out, in arid areas, plant growth is indeed more
sensitive to precipitation. We specified the location. Please see in line 65-68 in the revised
manuscript.

*“Fossil pollen reflects past vegetation communities that have specific temperature requirements in
both cold and warm seasons^{9,21-23}, and thus it is a potentially reliable proxy for seasonal
temperature reconstructions⁹ **except in arid regions where water availability may be the more
constraining factor.**”*

Methods [I read the methods before reading the results and discussion to avoid biases]

Data selection in such a large-scale study is always tricky and finding the right trade-off between
quality and quantity is most complex. While I generally agree with the procedure described here, I
can't help it but wanting to see the records from northeastern Russia (fig. SM1) included. This part of
the world is so under sampled that even low quality data might provide useful information, wouldn't
it?

**Responses:** Thank you for this constructive suggestion. We have relaxed the selection criteria for the
northern Eurasia region, i.e., retaining the records with resolution finer than 1000 years, instead of
400 years (see details in line 317-319 in the revised manuscript). The new selected sites can be seen
in the revised Supplementary Fig. 1 and Fig. R11 below in this document. Most of the sites in
northeastern Russia are still excluded in the selection. We further carefully checked the records from
this region. Unfortunately, most of them have no age controls, or have too short duration, or are
beyond the Holocene epoch. Therefore, this region is still under sampled and need to be further
studied in the future.

**Fig. R11:** Maps showing the selected (green circles) and all (smaller grey circles) pollen records.

L274-276: I am not sure I understand this argument. Winter temperature can be reconstructed from
the PFTs not because temp is included in the PFTs but because temp impact plants directly. The
inclusion of temp in the definition of PFTs is simply a reflection of that, and it doesn't give extra
strength to PFTs to reconstruct temperature. Or perhaps I am misunderstanding the argument here?
In any case, it needs to be revisited.

**Responses:** Thank you. We agree with your comments that winter temperature can be reconstructed
from PFTs data because the seasonal temperatures impact plant growth directly. Since this sentence
seems have no special information, it has been deleted in the revised manuscript.

L283-291: I fully agree with this paragraph, and it is something that is not sufficiently said in
publications. However, I am not sure that the terminology employed here with primary and
secondary variables helps. Winter and summer temperature are, in this case, two “primary variables”
and as the authors say, it is still complex to determine if one is stronger than the other. I would
therefore just stay away from this implicit variable ranking to avoid confusion.

**Responses:** Thank you. As you pointed out, the terminology of primary and secondary variables may
not be suitable, because both of winter and summer temperatures are key factors for plant growth.
Here we want to say that the relative contributions of summer and winter temperatures to vegetation
variation could be estimated and RDA analysis provides us a way. We have revised this paragraph.
Please see details in line 349-359 in the revised manuscript.

*“Past changes in both summer and winter temperatures have been widely reconstructed using pollen*
*data because of their direct effects on plant growth and distribution^{9,12}. However, challenges may*
*arise in the reconstruction of multiple climate variables, which might be biased if they show strong*
*covariance in the calibration data, but this covariance structure may vary over time⁷⁶⁻⁷⁸. Summer*
*temperature is frequently regarded as the dominant variable that drives changes in plant*
*composition, mainly according to the studies in the high latitudes^{77,78}. However, when considering*
*vegetation biogeography, cold tolerance and chilling requirements are also critical limiting factors*
*with respect to vegetation dynamics and distribution, and are perhaps more important than summer*
*temperatures^{22,23}. To date, it is still insufficient to examine the relative contributions of summer and*
*winter temperatures to vegetation changes.”*

L320: Why should covariance lead to ‘underestimation’ of the range, and not overestimation?

**Responses:** Thank you. It is indeed difficult to say ‘underestimation’ or ‘overestimation’. This
sentence may be misleading and lack of meaningful information, and thus we deleted it in the revised
manuscript.

L334-339: I think the authors use the wrong terminology here. It seems that they are referring to
spatial autocorrelation, but they wrote about covariance, which echoes the previous paragraph on the
covariance of the studied climate variables. Please adjust.

**Responses:** We carefully checked this sentence, and found no error in terminology use here. RDA
analysis in the previous paragraph indicates that the covariance of winter and summer temperatures
in PFT data is not very large. Here we want to say that MAT method may further reduce the
influence of covariance problem, relative to traditional transfer functions (TFs, such as weighted
averaging, WA).

The TFs use a constant function calibrated from the modern dataset to reconstruct the variable of
interest. However, the function may include the portion of covariance shared with other co-varying
variables. Thus, TFs are susceptible to the **covariance** problem when the covariance structure
between the variable of interest and others changes through time. Details are clarified in Juggins
(2013).

Spatial autocorrelation mainly leads to the overestimation of model performance in modern cross-
validation. MAT may be more susceptible than TFs (Telford and Birks, 2005, 2009), and we used the
*h*-block cross-validation to assess the influence of spatial autocorrelation. Please see details in line
430-431 in the revised manuscript. “*We also performed h-block cross-validation using the R package*
*palaeoSig⁸⁶ to assess the influence of spatial autocorrelation^{87,88}.”*

References:

Juggins, S. Quantitative reconstructions in palaeolimnology: new paradigm or sick science? *Quat.*
*Sci. Rev.* 64, 20–32 (2013).

Telford, R. J. & Birks, H. J. B. The secret assumption of transfer functions: problems with spatial
autocorrelation in evaluating model performance. *Quat. Sci. Rev.* 24, 2173–2179 (2005).

Telford, R. J. & Birks, H. J. B. Evaluation of transfer functions in spatially structured environments.
*Quat. Sci. Rev.* 28, 1309–1316 (2009).

The distance of 50km is rather small for the *h*-block analysis. Have the authors tried different
thresholds? Also, I am not convinced that a LOO analysis gives really useful results in this type of
studies. There are so many samples available to MAT that removing one is not going to have a big

effect. It is not wrong, and it can stay in the paper, but I do not think that it is really informative. The
HB is much more valuable.

**Responses:** Thank you. We used a distance of 50 km following the references of Williams et al.
(2008) and Salonen et al. (2014). However, the distance may be too small as you suggested. Under
your suggestion, we have done a series of experiments with changing distance- h . The results are
shown in the revised Supplementary Fig. 2 (a copy in Fig. R12 below in this document).

For all regions, the model performance (indicated by R^2) deteriorates gradually with the increasing h
values, after a sharp deterioration at the beginning (0-50 km). Correspondingly, the median squared
chord distance (SCD) increases, most markedly at the beginning, which indicates the analogue
situation worsens.

It is difficult to determine which h is the most suitable. Here we propose a solution. Firstly, we find
out the median SCD for the fossil samples during the Holocene. Secondly, we match the sample
SCD with the SCD curves of the changing- h experiments. The h value with the equivalent SCD is
considered to be the most suitable. As a result, the suitable h values for Europe, Eurasia, East Asia,
eastern North America, western North America, and Beringia are 100 km, 100 km, 400 km, 400 km,
200 km, and 200 km, respectively (see in Figure R2 and the revised Supplementary Fig. 2).

The corresponding model performance is shown in the revised Supplementary Table 2. Please see
details in line 433-438 in the revised manuscript.

*“The determination of h is a trade-off between reducing the influence of spatial autocorrelation and
the analogue quality^{77,78}. Here we proposed a method to find the suitable h . Firstly, we conducted a
series of cross-validation experiments with h increasing from 0 to 1000 km⁷⁸. Then we matched the
median SCD for the Holocene samples with the median SCD curves in the changing- h experiments.
The h value with the equivalent SCD was considered to be suitable (Supplementary Fig. 2).”*

With regard to LOO, it is a conventional method for evaluating model performance and is applied to
many studies (e.g., Davis et al., 2003; Marsicek et al., 2018). Therefore, we retain the LOO results
for comparisons with previous studies.

Fig. R12: Model performance under various h distance in h -block cross-validation. The horizontal dotted lines
 indicate the median SCD for the fossil samples during the Holocene. The vertical dotted lines indicate the
 selected suitable h distance.

References:

Williams, J. W. & Shuman, B. Obtaining accurate and precise environmental reconstructions from
 the modern analog technique and North American surface pollen dataset. *Quat. Sci. Rev.* 27, 669–
 687 (2008).

Salonen, J. S. et al. Reconstructing palaeoclimatic variables from fossil pollen using boosted
 regression trees: comparison and synthesis with other quantitative reconstruction methods. *Quat. Sci.*
 *Rev.* 88, 69–81 (2014).

Davis, B. A. S., Brewer, S., Stevenson, A. C. & Guiot, J. The temperature of Europe during the
 Holocene reconstructed from pollen data. *Quat. Sci. Rev.* 22, 1701–1716 (2003).

Marsicek, J., Shuman, B. N., Bartlein, P. J., Shafer, S. L. & Brewer, S. Reconciling divergent trends
and millennial variations in Holocene temperatures. *Nature* 554, 92–96 (2018).

L377: Why not use the same reference for all the records. This is confusing.

**Responses:** Thank you. We use the same reference (i.e., 1000 yr) for all records now.

L378-381: I cannot understand what the authors did here. It needs some rephrasing and a bit more
details.

**Responses:** Thank you for the suggestion. We have rephrased this paragraph. More details have been
added in line 478-491 in the revised manuscript.

*“Following Marsicek et al.⁹, all of the significant reconstructions after isostatic correction were*
*converted to anomalies using the mean of the latest 1000 years, linearly interpolated to 200-year*
*bins, averaged onto a 2° × 2° grid, and finally generated the hemispheric and regional composites of*
*Holocene seasonal temperatures. The errors produced in courses of reconstruction, temporal*
*interpolation, and stacking were fully considered to estimate the uncertainties in the composites. The*
*sample-specific errors in reconstructions were estimated via bootstrapping the training-set samples*
*for each fossil sample using 100 iterations, and then combining the bootstrap-derived error with the*
*model RMSEP in the leave-one-out cross-validation⁸⁷. In temporal interpolation for each site, we*
*used a Monte Carlo method to randomly sample a normal distribution of the sample-specific errors,*
*and generated an ensemble of 1000 time series to interpolate at 200-year bins separately, from*
*which the median and standard error were derived. Similarly, a 1000-iteration Monte Carlo*
*sampling of the interpolating series with errors was implemented to calculate the median and 95%*
*uncertainty intervals of the final composites.”*

Did the author correct for the isostatic rebound of northern Europe and Canada? Across the
Holocene, the changes in elevation will have a significant effect on the reconstructed temperatures
without being driven by climate change per se. See Mauri et al. (2015) [listed in the bibliography
already] for one solution to the problem. There are probably other ways. Independently of the
method used, accounting for this effect is quite important.

**Responses:** Thank you for this suggestion. We take your advice and correct for the isostatic
adjustment at Europe and North America. Please see details in line 463-475 in the revised manuscript

and in the revised Supplementary Fig. 3 (a copy Fig. R13 below in this document). In general, the
isostatic correction has minor influence on the reconstructed temperature trends over the Northern
Hemisphere landmass.

*“The gradual decay of last glacial ice sheets over northern Europe and North America had led to the*
*isostatic adjustment of continental surface as the surface load was removed⁹³, which may also have*
*important effect on the interpretation of reconstructed temperatures during the Holocene¹⁴.*
*Following Mauri et al.¹⁴, we calibrated the reconstructed temperatures through subtracting the*
*topography-induced temperature changes. In order to calculate this component, we first used the*
*output of topographic anomalies from the latest ICE-7G_NA (VM7) model⁹³ (1 degree spatial and*
*500-year temporal resolution) to obtain the history of topographic changes for each site by spatial*
*and temporal interpretations using inverse distance weighted and linear methods, respectively. We*
*then calculated the topography-induced temperature changes using the present-day locally (within*
*300 km) topographic lapse rate of temperatures, which were calculated using the CRU climate*
*dataset⁶⁸. After the isostatic correction, the composite temperatures induced by climate change*
*became a bit lower, but with negligible amplitude of variation (Supplementary Fig. 3).”*

Fig. R13: Comparison between reconstructions with (red) and without (blue) isostatic correction

Seasonal Temperature Evolution

The colour scale of fig. 2 should be improved. At the moment, it looks like the vast majority of the
points are coloured within either < -0.5 or $> +0.5$. Expanding this a bit to gain better insight of the
spatial patterns is required.

**Responses:** Thank you. As for your concern, we have expanded all the colour bar (i.e., 0.7, 0.4, 0.1,
0). Please see details in the revised Fig.2 and Supplementary Figs. 5, 6, 8 and 9. In this study, we
focus on the Holocene temperature trends, and thus we show the correlation coefficient between the
reconstructed series and PC1 for EOF patterns. The value >0.7 indicates strong correlation; 0.7-0.4,
moderate correlation; 0.4-0.1, weak correlation; and 0.1-0, negligible correlation (Schober et al.,
2018).

References:

Schober, P., Boer, C. & Schwarte, L. A. Correlation Coefficients: Appropriate Use and
Interpretation. *Anesth. Analg.* 126, 1763–1768 (2018).

Model-data comparison

I think it would be important to note the scaling differences between the data and the models.
Changes in TRACE are 2 to 3 times larger than the changes reconstructed from pollen. Might that be
a bias in the data, or in the stacking approach?

**Responses:** Thank you for this comment. In the previous version of reconstruction, the changes in
reconstructions are indeed much smaller than simulations. It may partially result from the possibility
that our previous version includes a portion of records which are not robust. In the new version, we
implemented significant test to select the potentially robust reconstructions (following the suggestion
of Reviewer #1, see details in line 447-462 in the revised manuscript). And the amplitudes of our
new composites of annual and seasonal temperatures have been similar with the ones from
simulations (see details in the revised Fig. 3).

For example, the amplitude of summer temperature during the Holocene is ~ 1.4 °C for data, and
~ 1.1 °C for model. From 11-7 ka BP, the amplitude of annual (winter) temperature is ~ 2.4 °C
(~ 4.1 °C) for data, and ~ 2.5 °C (~ 3.7 °C) for model. However, divergent trends exist from 7 to 0 ka
BP, when our reconstructed annual (winter) temperatures decreased by ~ 0.5 °C (~ 0.7 °C), whereas
the simulated annual (winter) temperatures increased by ~ 0.3 °C (~ 1.2 °C). The divergence in winter

temperature leads to the larger amplitude of Holocene seasonality variation in model (~5.0 °C) than
in reconstruction (~2.7 °C). We have discussed the divergence in details in line 237-295 in the
revised manuscript, and it may reflect the uncertainties in models associated with feedback
processes.

We have added these comparisons in line 183-191 in the revised manuscript.

*“For winter and annual temperatures, the warming from ~11 to 7 ka BP is present in both the*
*reconstructions and model results with similar magnitudes (~2.4 and 2.5 °C in annual, and ~4.1 and*
*3.7 °C in winter for reconstructed and simulated temperatures respectively; Fig. 3a, c), although*
*clear differences exist in western North America and southern Asia where the models show a*
*coherent warming trend, whereas the reconstructions show an HTM phase (Supplementary Figs. 4*
*and 8). Pronounced discrepancies, however, occur after ~7 ka BP, when the simulated temperature*
*continues to rise until the present (~0.3 and 1.2 °C in annual and winter temperatures, respectively),*
*unlike the pervasive cooling trends in the reconstructions (Fig. 3a, c and Supplementary Fig. 9).”*

Drivers of Holocene temperature change

L208-211: I do not understand this argument. Could you explain in greater details why
reconstructing different trends in North America and Europe is a validation of an absence of warm-
season bias? The trends might well be truly different for the two regions and the proxies still biased
towards summer temp (hence a higher capacity to reconstruct small changes and eventually
reconstructing divergent trends between regions). Please refine your argument here.

**Responses:** Thank you for pointing out this. We are sorry that we didn't clearly express our idea
here. We don't intend to say the different trends between North America and Europe. Here we intend
to express that our reconstructed divergent trends between winter and summer temperatures in these
regions can indicate a good performance in extracting seasonal signals. We have rephrased this
paragraph. Please see details in line 249-254 in the revised manuscript.

*“Another potential reason for the reconstructed mid- to late Holocene winter and annual cooling*
*trend is that it may represent the warm-season temperature, as was suggested for the marine-*
*dominant multiproxy reconstructions^{5,6}. However, our reconstructions show a good performance in*
*extracting seasonal signals. In regions such as western North America, and western and southern*
*Europe, divergent trends between seasonal temperatures were well reconstructed (Supplementary*
*Figs. 4–6).”*

Reviewer #3 (Remarks to the Author):

Reviewer report

NCOMMS-21-43673-T

Holocene seasonal temperature evolution over the Northern Hemisphere landmass

Key results

Accurate records of timing, magnitude, spatial and seasonal aspects of Holocene temperature change
are valuable for understanding the relative role of different external and internal climate forcings and
their Earth system feedbacks. Zhang et al. provide an extensive stacked reconstruction of Holocene
northern hemisphere temperature trends, with focus on the role of seasonality, based on fossil pollen
data and methodologically carefully selected and well-validated approaches. They further compare
the results to TraCE-21ka model simulation. The results demonstrate, supporting previous
reconstructions, that early Holocene warming in both summer and winter temperatures was followed
by a thermal maximum around 7 ka and cooling until the pre-industrial era. Prominent and
decreasing seasonality (difference between summer and winter temperatures) in the early Holocene
was settled by ca. 7 ka. The results, based on the PFT (plant function type) approach, provide an
important contribution to the topical discussion of global and large-scale temperature development
over the Holocene. Importantly, proxy-based evidence of winter temperature changes support
previous proxy-based consensus of late Holocene cooling (see however, recent work by Osman et al.
2021) challenging the capability of present climate models (that show late Holocene warming) in
simulating Earth system feedbacks.

**Responses:** We are grateful for your encouraging comments.

Validity

Zhang et al. based their findings on geographically and numerically extensive combination of pollen
databases. They carefully evaluate and filter the original data (radiocarbon dates, pollen counts,
resolution), their own approaches and results (variation partitioning and RDA, comparison with

“individual proxies” and model results over data sites; reasoning for choosing MAT for
reconstruction method; Monte Carlo and bootstrap resampling for error estimation).

**Responses:** Thank you again for your comments.

Significance

There has been a recent increase in global and large-scale Holocene climate reconstructions based on
the recent buildup, management and use of extensive open data banks. These are grounded on a
mixture or set of proxy sources, some including both or only marine/terrestrial evidence, some on
transfer functions (microfossil assemblage based) and geochemical methods (leaf waxes, alkenones,
GDGTs), and some on a set of proxy type. The present contribution by Zhang et al. is a very
welcome addition, contributing to the very topical discussion on Holocene climate trends and
relevant forcings and system feedbacks. Zhang et al. work is important in particular because:

- It is based on an impressive dataset, with a more extensive distribution of sites compared to
previous sets (Marcott et al. 2013; Kaufman et al. 2020), in particular from East Asia

- It provides a sound continental-only view, based on one climatically very sound proxy, i.e. pollen,
which is then treated carefully through biomization and plant functional types PFTs (there are marine
only stacks, but not such large-scale continental only stacks)

- It provides support on proxy (pollen) ability to reconstruct seasonality, challenging model
understanding on Earth system feedbacks. This provides direction for future work with great
relevance to present day climate science.

**Responses:** Thank you again for your encouraging comments. We will do our best to improve our
manuscript.

Data and methodology

The data quality and validity of used approaches, including MAT, RDA, biomization and stacking,
are very high and reasonably well explained and presented in Extended datasets. The selection of
PFT over pollen taxa for reconstruction is well described, however see my comments elsewhere on
the depth of mechanistic reasoning and discussion.

**Responses:** Thank you again for your comments.

Analytical approach

The analytical approach from database filtering, use of biomization of pollen data, buildup of state-
of-the-art age-depth models (huge effort!), testing of significance of variables and variation
partitioning (RDA), as well as testing of Modern Analogue (MAT) based reconstructions (basic
performance statistics, leave-one-out, h-block) and error evaluation in the reconstruction stacking
procedure. However, I did not find the results of most tests in the files submitted. If this is indeed the
case, please provide them in the Supplementaries, regarding MAT testing e.g. closest analogue
locations/statistics.

**Responses:** Thank you. The testing of significance of variables using RDA is provided in the revised
Supplementary Table 1 (also see a copy Table R2 below in this document). The testing of MAT
model has been provided in the revised Supplementary Table 2 (also see a copy Table R3 below in
this document).

Table R2. Contributions of MTCO and MTWA to the variation of modern PFT score data based on
RDA. *R* is Pearson correlation coefficient for the MTCO and MTWA in the calibration datasets.
λ_1/λ_2 , the ratio of the eigenvalues of the constrained and the first unconstrained axes in single-
variable RDA runs, represents the relative explanatory power of the selected variable. *Explained* and
*Unique* are the whole and unique proportions of total variation explained by each variable in the
partial RDA. Correspondingly, the whole and shared proportions for both variables are displayed in
the following row (*Common*). The significance (*p* value) of each variable with respect to variation
explaining was assessed using Monte Carlo permutation tests (1000 permutations). The RDA was
run separately in the six regions used for the PFT calculations.

Region	Variable	R	λ_1/λ_2	Explained (%)	Unique (%)	Covariance prop. (%)	P
Europe	MTCO		0.92	14.8	7.9	46.5	0.001
	MTWA		0.60	11.3	4.4	60.8	0.001
	Common	0.67		19.2	6.9	35.8	
Eurasia	MTCO		0.51	9.9	5.8	40.9	0.001
	MTWA		0.31	6.3	2.3	64.1	0.001
	Common	0.57		12.1	4.0	33.3	
East Asia	MTCO		0.68	13.4	10.1	24.8	0.001

	MTWA		0.37	7.9	4.6	42.0	0.001
	Common	0.54		18.0	3.3	18.5	
Canada and the eastern United States	MTCO		1.15	20.4	5.4	73.7	0.001
	MTWA		1.02	20.5	5.5	73.1	0.001
	Common	0.76		25.9	15	58.0	
western United States	MTCO		0.82	17.9	10.2	43.2	0.001
	MTWA		0.87	16.5	8.8	46.9	0.001
	Common	0.64		26.7	7.7	29.0	
Beringia	MTCO		0.57	13.5	13.3	1.5	0.001
	MTWA		0.36	10.4	10.2	2.0	0.001
	Common	0.35		23.7	0.2	0.9	

Table R3. MAT performance for temperature reconstructions. For each climate variable, the
coefficient of determination (R^2) and the root-mean-square error of prediction (RMSEP, °C) were
calculated by leave-one-out (LOO) and h -block (HB) cross validation. The number of modern
analogues was determined from the lowest RMSEP among the 5–7 closest analogues. The distance
(h , km) for HB was determined by the squared chord distance through a series of experiments (see
Methods and Supplementary Fig. 2).

Region	Variable	R^2 , LOO	RMSEP , LOO	R^2 , HB	RMSE P, HB	No. analogue	Thre- shold	h
Europe	ANNT	0.85	2.0	0.70	2.8	6		
	MTCO	0.87	2.4	0.74	3.3	6	0.2	100
	MTWA	0.81	2.0	0.61	2.9	7		
Eurasia	ANNT	0.85	3.1	0.74	4.0	7		
	MTCO	0.85	4.5	0.72	6.3	7	0.3	100
	MTWA	0.69	2.8	0.54	3.5	6		
East Asia	ANNT	0.80	3.4	0.62	4.8	7		
	MTCO	0.86	4.2	0.65	6.5	5	0.5	400
	MTWA	0.68	3.7	0.42	5.0	7		
Canada and the eastern United States	ANNT	0.91	2.5	0.76	4.0	7		
	MTCO	0.89	3.8	0.70	6.2	7	0.3	400
	MTWA	0.89	1.9	0.76	2.8	7		
western United States	ANNT	0.91	2.2	0.81	3.3	7		
	MTCO	0.93	2.8	0.85	4.3	7	0.3	200
	MTWA	0.85	2.1	0.68	3.1	7		
Beringia	ANNT	0.80	2.4	0.63	3.3	7		
	MTCO	0.75	3.7	0.50	5.3	7	0.3	200
	MTWA	0.84	1.5	0.72	2.0	7		

Based on the general knowledge and testing, MAT is probably a good choice for the present large-
scale dataset; however, it would have been interesting to see whether other methods such as machine
learning or Bayesian based, would have given similar results. I understand such effort might not be

feasible for the present manuscript, and as stated above, I consider the approach and evaluation
carefully considered. However, I would like to see some discussion on the possible influence of the
choice of reconstruction method on the results.

**Responses:** Thank you. We take your suggestions and conduct the reconstructions using two other
methods based on PFT score data, namely weighted averaging-partial least squares (WA-PLS) and
boosted regression tree (BRT, a machine learning method). Please see the descriptions about the two
methods in line 494-526 in the revised manuscript, the modern validation results in a separate file -
Supplementary Data 3 (a copy Table R4 below in this document), and the reconstructed Northern
Hemisphere temperatures in Supplementary Fig. 3 (a copy Fig. R14 below in this document).

The temperature trends reconstructed by MAT, WA-PLS and BRT are generally consistent with each
other, with an early to mid- Holocene warming and mid- to late Holocene cooling trend (Fig. R14
below in this document), supporting the robustness of our reconstructions. Inevitably, there is some
minor difference among methods. The WA-PLS-based temperatures has a most pronounced peak at
~7 ka BP, while winter temperature using BRT has a warmer early Holocene than the other twos.

*“To further evaluate our MAT-based Holocene temperature trends, we conducted reconstructions*
*using two other common methods based on PFT score data, namely a classical transfer function –*
*WA-PLS⁸³ and a newly-developed machine-learning approach – boosted regression tree (BRT)^{77,78}.*
*WA-PLS and BRT were implemented with the R package Rioja^{78,87} and gbm^{78,94}, respectively. We*
*also generated reconstructions using the MAT method based on pollen taxa data. 60 and 40 taxa*
*were selected for North America and Europe, respectively, following Marsicek et al.⁹, while 56 and*
*74 common taxa from the taxa–PFT matrixes^{71,72} were used for northern Asia and East Asia regions,*
*respectively (Supplementary Data 2).*

*The models’ performance indicated by leave-one-out and h-block cross-validations can be seen in*
*Supplementary Data 3. Overall, MAT performs the best in leave-one-out cross-validation for all*
*variables and all regions, and performs similar with BRT in h-block cross-validation, while WA-PLS*
*does the worst. These results support the relative advantages of MAT for large scale*
*reconstructions^{9,36}. MAT using PFT scores has comparable performance with MAT using taxa data*
*in Europe and Asia continents, while the latter performs better in North America, which may benefit*
*from the more degrees of freedom in model construction. Additionally, all the methods indicate a*
*generally better performance for MTCO than MTWA with a higher R^2 (Supplementary Data 3).*

*Following the same procedure for MAT, we implemented significance testing, isostatic correction,*
*temporal interpolation and stacking for the new reconstructions. For MAT based on taxa data, 816*

records are retained for ANNT (62%), 781 for MTCO (60%), and 821 for MTWA (63%) in the
 significance-test step. For WA-PLS, the count of significant records is 308 for ANNT (23%), 325 for
 MTCO (25%) and 286 for MTWA (22%). Fewer than 200 records, however, yielded significant
 reconstructions for BRT (177 for ANNT, 153 for MTCO, and 149 for MTWA), and thus we skipped
 the significance selection for BRT to ensure the spatial coverage. This extremely low ratio (<15%) of
 significant reconstructions warns that the palaeoSig approach⁸⁸ may be too conservative, as
 discussed above.

The Holocene temperature trends over the NH landmass reconstructed using MAT (based on both
 PFTs and taxa data), WA-PLS and BRT methods are generally consistent with each other, with an
 early to mid-Holocene warming and thereafter cooling trend for both annual and seasonal
 temperatures (Supplementary Fig. 3), further supporting the robustness of our reconstructions.
 Inevitably, there is some minor difference among methods. The temperatures reconstructed using
 WA-PLS has a most pronounced peak at ~7 ka BP, while winter temperature using BRT has a
 warmer early Holocene than the other twos.”

 Fig. R14: Comparison of Holocene temperature trends over the Northern Hemisphere reconstructed by MAT,
 BRT and WA-PLS.

Table R4. Model performance of MAT, BRT and WA-PLS for temperature reconstructions.

Region	Variable	R ² , LOO			RMSEP, LOO			R ² , HB			RMSEP, HB		
		MAT	BRT	WA-PLS	MAT	BRT	WA-PLS	MAT	BRT	WA-PLS	MAT	BRT	WA-PLS
Europe	ANNT	0.85	0.80	0.67	2.0	2.3	2.9	0.70	0.72	0.65	2.8	2.7	3.0
	MTCO	0.87	0.82	0.67	2.4	2.7	3.7	0.74	0.75	0.65	3.3	3.2	3.8
	MTWA	0.81	0.75	0.60	2.0	2.3	2.9	0.61	0.64	0.57	2.9	2.7	3.0
Eurasia	ANNT	0.85	0.80	0.65	3.1	3.5	4.6	0.74	0.72	0.61	4.0	4.2	4.9
	MTCO	0.85	0.79	0.62	4.5	5.4	7.2	0.72	0.69	0.57	6.3	6.6	7.7
	MTWA	0.69	0.62	0.45	2.8	3.1	3.8	0.54	0.51	0.40	3.5	3.6	3.9
East Asia	ANNT	0.80	0.78	0.69	3.4	3.6	4.3	0.62	0.64	0.64	4.8	4.7	4.6
	MTCO	0.86	0.83	0.75	4.2	4.5	5.5	0.65	0.66	0.69	6.5	6.4	6.1
	MTWA	0.68	0.65	0.52	3.7	3.8	4.5	0.42	0.45	0.45	5.0	4.8	4.8
Canada and the eastern United States	ANNT	0.91	0.88	0.79	2.5	2.8	3.8	0.76	0.77	0.72	4.0	3.9	4.3
	MTCO	0.89	0.86	0.73	3.8	4.2	5.8	0.70	0.71	0.64	6.2	6.0	6.7
	MTWA	0.89	0.86	0.75	1.9	2.1	2.8	0.76	0.77	0.71	2.8	2.7	3.0
western United States	ANNT	0.91	0.88	0.79	2.2	2.6	3.5	0.81	0.81	0.76	3.3	3.3	3.7
	MTCO	0.93	0.90	0.82	2.8	3.4	4.7	0.85	0.85	0.80	4.3	4.3	5.0
	MTWA	0.85	0.80	0.69	2.1	2.4	3.0	0.68	0.70	0.65	3.1	3.0	3.2
Beringia	ANNT	0.80	0.76	0.61	2.4	2.6	3.6	0.63	0.63	0.56	3.3	3.3	3.6
	MTCO	0.75	0.72	0.52	3.7	4.0	5.2	0.50	0.53	0.44	5.3	5.2	5.6
	MTWA	0.84	0.81	0.64	1.5	1.6	2.3	0.72	0.72	0.60	2.0	2.0	2.4

Clarity and context

The manuscript is very easy to read, logically presented and broad audience friendly. There is
 sufficient context and appropriate consideration of previous work throughout the text. However, the
 discussion is simple to the point that I find it superficial in places, which gives a (false?) impression
 that the authors have not considered the context and interpretations fully. The authors use appropriate
 references, but the reader is not going to go searching for explanations in the original publications;
 the authors are supposed to state their own assessment and mechanistic interpretations. A few
 examples:

**Responses:** Thank you for this comment. We have tried our best to extend our interpretations. Please
 see point-by-point responses below.

Lines 38-40 While it is a good idea to start with a general sentence, there could be a bit more depth
 to this; e.g. with respect to importance of forcings and feedbacks of the system.

**Responses:** Thank you. We revised this sentence. Please see details in line 41-43 in the revised
manuscript. *“Knowledge of past climatic conditions is essential to improve our understanding of the*
*climate system with respect to the climate forcings and feedbacks, and to better constrain future*
*climate projection¹.”*

Lines 140-142 This is a very interesting result and put into context of previous work; however the
reasons are not discussed from climate forcing/feedback perspective here or in the later section on
drivers.

**Responses:** Thank you. The Holocene winter warming trend at northern Asia is a very interesting
topic. We have been thinking about the mechanism behind. Our opinion is that this pattern of
temperature change may reflect the influence of Arctic sea ice change. Using multi-model
simulations, Park et al. (2018; 2019) demonstrated that the Arctic sea ice loss due to the high summer
insolation during the mid-Holocene warmed the North America especially in winter season (~ 2 °C),
but cooled northern Asia (~ 0.6 °C), via processes associated with Arctic amplification, weakening of
mid-high latitude westerlies and northern Asia cooling (see Fig. R15, a copy from Park et al. 2018).
Our reconstructed winter warming trend at northern Asia since the mid-Holocene may reflect the
combined impact of the increase of winter insolation forcing and the increasing Arctic sea-ice
coverage (Werner et al., 2016; Berben et al., 2017) induced by decrease of boreal summer insolation
(Park et al., 2018).

The impact of Arctic sea ice loss can also be seen in the climate changes of recent decades. The
meteorological observations show that severe cold winters happened in mid-latitude Eurasia in recent
decades despite global warming, so called warm Arctic–cold continents pattern (e.g., Overland et al.,
2011; Cohen et al., 2014; Mori et al., 2014; Kug et al., 2015; Cohen et al., 2020). Simulations reveal
that the positive feedbacks between Arctic surface temperature and sea-ice loss in Barents–Kara Sea
region will reduce the equator-to-pole surface temperature gradient, weaken midlatitude westerly
winds, and thus promote cold winters in mid-latitude Eurasia (eg., Mori et al., 2014; Kug et al., 2015;
Mori et al., 2019; Cohen et al., 2020), which is similar with the situation during the mid-Holocene
(Park et al. 2018). This suggests that the Arctic sea ice may have important impact on the Eurasia
climate at different time scales.

We have added these discussions in line 276-286 in the revised manuscript.

*“Sea ice feedback is another important process that can influence Holocene temperature evolution⁴⁴⁻*
*⁴⁷. The models with stronger sea-ice responses to summer insolation changes tend to generate a*

*year-round warmer mid-Holocene in the NH⁴⁵, and the sea ice loss-induced winter warming can*
*almost counteract the cooling induced by the lower winter insolation in the mid-high latitudes⁴⁶.*
*Models also indicated an asymmetric pattern of winter temperature responses to Arctic sea ice loss*
*associated with the weakening of mid-high latitude westerlies in the mid-Holocene: the northern Asia*
*cooled, while the Arctic region and North America warmed strongly⁴⁶. Our reconstructions support*
*the importance of sea-ice feedbacks by showing a consistent spatial pattern that there is a warming*
*trend in northern Asia and overall cooling trend in North America after ~7 ka BP (Fig. 2 and*
*Supplementary Fig. 4), when the Arctic sea-ice coverage increased under the forcing of decreasing*
*summer insolation^{50,51}.*”

Fig. R15: Wintertime temperature responses to Arctic sea ice loss during the mid-Holocene (a copy from Park
et al. 2018).

References:

Park, H. et al. The impact of Arctic sea ice loss on mid-Holocene climate. Nat. Commun. 9, 4571
(2018).
Park, H. S., Kim, S. J., Stewart, A. L., Son, S. W. & Seo, K. H. Mid-Holocene Northern Hemisphere
warming driven by Arctic amplification. Sci. Adv. 5, x8203 (2019).
Werner, K. et al. Holocene sea subsurface and surface water masses in the Fram Strait –
Comparisons of temperature and sea-ice reconstructions. Quat. Sci. Rev. 147, 194–209 (2016).

Berben, S. M. P., Husum, K., Navarro-Rodriguez, A., Belt, S. T. & Aagaard-Sørensen, S. Semi-
quantitative reconstruction of early to late Holocene spring and summer sea ice conditions in the
northern Barents Sea. *J. Quat. Sci.* 32, 587–603 (2017).

Overland, J. E., Wood, K. R. & Wang, M. Warm Arctic-cold continents: climate impacts of the
newly open Arctic Sea. *Polar Res.* 30, 15714–15787 (2011).

Cohen, J. et al. Recent Arctic amplification and extreme mid-latitude weather. *Nat. Geosci.* 7, 627–
637 (2014).

Mori, M., Watanabe, M., Shiogama, H., Inoue, J. & Kimoto, M. Robust Arctic sea-ice influence on
the frequent Eurasian cold winters in past decades. *Nat. Geosci.* 7, 869–873 (2014).

Kug, J. et al. Two distinct influences of Arctic warming on cold winters over North America and
East Asia. *Nat. Geosci.* 8, 759–762 (2015).

Mori, M., Kosaka, Y., Watanabe, M., Nakamura, H. & Kimoto, M. A reconciled estimate of the
influence of Arctic sea-ice loss on recent Eurasian cooling. *Nat. Clim. Change* 9, 123–129
(2019).

Cohen, J. et al. Divergent consensus on Arctic amplification influence on midlatitude severe winter
weather. *Nat. Clim. Change* 10, 20–29 (2020).

Lines 186-190 The authors provide no mechanistic evidence/explanation why they think the NH
seasonal temperature patterns and trends would have been controlled by boreal summer insolation,
modulated by ice sheets.

**Responses:** Thank you. We have reframed the paragraphs to make the explanation clearer. Please
see details in line 205-236 in the revised manuscript (also a copy below). We also explained the
reason why summer insolation could be important for winter and annual temperature changes, not
only for summer temperature, through feedbacks such as vegetation and sea ice, etc. (line 261-295 in
the revised manuscript).

*“Our reconstructions show that Holocene temperature evolution over the NH landmass in both*
*summer and winter seasons was characterized by a warming trend during ~11–7 ka BP, followed by*
*a cooling trend after ~7 ka BP. The early to mid-Holocene warming occurred mainly in eastern*
*North America and Europe; this was successfully simulated by the models (Supplementary Fig. 8).*
*Single forcing experiments from TraCE-21ka indicate that this warming was associated with the*

*retreat of the NH ice sheets (Fig. 4), which can strongly influence temperatures via direct depression*
*and by its effect on surface albedo, meltwater flux, and atmospheric circulation*^{5,34}.

*Besides the ice-sheet forcing, single forcing experiments show that orbital-induced seasonal*
*insolation changes play a critical role in seasonal temperature evolutions, resulting in summer*
*cooling and winter warming trends, and thus a long-term decrease in temperature seasonality during*
*the Holocene (Fig. 4b–d). Our reconstructions support the dominant impact of seasonal, especially*
*summer insolation. In regions far from or upwind of the ice sheets, such as southern Asia and*
*western North America, both annual and seasonal temperatures reached a peak at the early*
*Holocene (Supplementary Fig. 4), which suggests the reduced influence of ice sheets and may reflect*
*the high summer insolation then. After ~7 ka BP, when the Laurentide ice sheet had almost*
*disappeared*³⁵, *widespread cooling is evident in the NH during both summer and winter seasons in*
*the reconstructions, which is consistent with the decreasing summer insolation. Therefore, the*
*reconstructed NH seasonal temperature patterns during the Holocene may have been controlled by*
*boreal summer insolation and modulated by NH ice sheets.*

*Winter insolation can also play an important role in the reconstructed winter temperature changes,*
*which is evident in the TraCE-21ka simulation (Fig. 4c). Single forcing experiments indicate that the*
*reconstructed amplitude of the winter temperature increase from ~11 to 7 ka BP (~4.1 °C) is unlikely*
*to be generated by ice-sheet forcing alone (~1.0 °C), and the rising winter insolation leads to*
*another ~1.0 °C of warming (Fig. 4c). However, the rising winter insolation could not be the*
*dominant factor for the mid- to late Holocene winter cooling in reconstruction (Fig. 4c). This winter*
*cooling trend may result from the strongly influence of summer insolation via feedback processes,*
*which we will discuss below. As the contribution of other forcing factors is similar during winter and*
*summer, the reduced insolation seasonality resulted in a long-term decrease in temperature*
*seasonality during the Holocene, which is consistently shown in the reconstructions and simulations*
*(Fig. 4d).”*

Lines 202-203 What are these non-climatic factors?

**Responses:** We have clarified these non-climatic factors. Please see details in line 242-244 in the
revised manuscript. “In addition to temperature, precipitation^{21,22,36} and several non-climatic
factors^{36,37} (e.g., soil, topography, fire and land use) can influence plant growth”

Line 208 What do you mean by warm season bias, how it is expressed and why it might be a
“missing” bias

**Responses:** The original expression is too simple and confusing. We rephrased this sentence. Please
see details in line 249-251 in the revised manuscript. *“Another potential reason for the reconstructed*
*mid- to late Holocene winter and annual cooling trends is that they may represent the warm-season*
*temperatures, as was suggested for the marine-dominant multiproxy reconstructions^{5,6}.”*

Lines 213-214 Why is winter temperature more critical than summer temperature or precipitation
(over NH landmass)? Is this “critical influence” true across the NH and all vegetation types?

**Responses:** The contribution of summer and winter temperatures to vegetation changes is
complicated. We have estimated it in detail at the Methods section, using Redundancy Analysis. In
general, the winter temperature explained evidently more vegetation variation than summer
temperature in almost all regions, e.g., Europe, Asia and western North America, while their
contribution is similar in eastern North America. Please see details in line 389-394 in the revised
manuscript and in the revised Supplementary Table 1. Certainly, downscaling to local scale, the
situation is much more complicated. For example, summer temperature may contribute more to the
vegetation variation in northern Europe (Salonen et al., 2014). However, in this study, we mainly
focus on the large-scale climate reconstructions.

We rephrased this sentence for better understanding. Please see details in line 255-260 in the revised
manuscript.

*“In addition, our reconstructions could be more robust for winter than for summer temperatures,*
*because of its more contribution to vegetation variation according to Redundancy Analysis, and the*
*better performance of the calibration for winter temperature indicated by cross-validations (see*
*details in Methods, Supplementary Tables 1 and 2, and Supplementary Data 3).”*

**References:**

Salonen, J. S. et al. Reconstructing palaeoclimatic variables from fossil pollen using boosted
regression trees: comparison and synthesis with other quantitative reconstruction methods.
Quat. Sci. Rev. 88, 69–81 (2014).

Suggested improvements

Key suggestions:

- See comments below on the lack of depth and mechanistic approach in discussion that should be
improved throughout the manuscript.

**Responses:** Thank you. We have tried our best to improve our manuscript following your
suggestions. Please see point-by-point responses below.

- Please include Osman et al. 2021 Nature in the discussion and reference list

**Responses:** Thank you. We have cited and discussed this recent and wonderful research of Osman et
al. Please see details in line 48, 49, 62, 64, 94, 97, 198, 295 and 580-581 in the revised manuscript.

Other minor suggestions, see also comments in other sections of review report:

Lines 64-66 Explain here or elsewhere, how your choice of method (MAT) might affect the results,
as compared to if you chose WA-PLS, machine learning or Bayesian methods. There is work that
combines different methods, see e.g. [Salonen, J.S., Korpela, M., Williams, J.W. et al. Machine-
learning based reconstructions of primary and secondary climate variables from North American and
European fossil pollen data. Sci Rep 9, 15805 (2019). <https://doi.org/10.1038/s41598-019-52293-4>]

**Responses:** Thank you. We have conducted reconstructions using WA-PLS and a machine learning
method (boosted regression tree, BRT) to compare with MAT. Please see details in line 494-526 in
the revised manuscript and in the revised Supplementary Fig. 3. Details can also be seen above in
this document (line 1065-1112).

Lines 252-253 <200 pollen count (the number of pollen counted, not the number of grains present)

**Responses:** Thank you. We have modified this part. Please see in line 319 in the revised manuscript.
“(5) *Samples with <200 pollen count were excluded.*”

Line 268 This is really not true; while some taxa such as aquatic species can be excluded, most taxa-
based quantitative reconstructions used in published regional and global compilation reconstructions
have used most available (not few indicator) taxa. Please revise.

**Responses:** Thank you for this comment. We deleted the words “Selected indicator”. Please see
details in line 334 in the revised manuscript. *“Pollen taxa are commonly used in the reconstruction*
*procedure⁹.”*

Lines 305-306 Is around 20% explanation of the total variance a lot? How about the rest of the 80%
of variance that influence the reconstructions? Please elaborate a bit on this for example by providing
context from other studies or methodology.

**Responses:** Thank you. As was discussed in our manuscript (line 242-248 in the revised
manuscript), vegetation compositions are influenced by multiple factors, including climatic (e.g.,
temperature and precipitation) and non-climatic factors (e.g., soil, topography, fire and land use). The
large size and coverage of our dataset further increased the complexity. ~20% explanation is quite
good for the large dataset. For comparison, Salonen et al. (2014) reported an upmost 8.1%
contribution of the total variation of pollen assemblages for summer temperature (3.9% for winter
temperature and 6.5% for water balance) in northern Europe (a smaller dataset than ours). They
attributed this level of explanation to the complexity of pollen assemblages and large size of dataset
(Salonen et al., 2014).

We have added some explanation for this. Please see details in line 372-373 in the revised
manuscript. *“Together, they explain more than 20% (23.7% to 26.7%) of the total variance in North*
*America, 18.0% to 19.2% in East Asia and Europe, and 12.1% in Eurasia. The proportions of*
*explanation are fairly high in consideration of the vegetation complexity in large datasets⁷⁷.”*

**References:**

Salonen, J. S. et al. Reconstructing palaeoclimatic variables from fossil pollen using boosted
regression trees: comparison and synthesis with other quantitative reconstruction methods.
*Quat. Sci. Rev.* 88, 69–81 (2014).

Lines 673-64 I would not amplify the reconstruction seasonality, why would you do this? If you do
so, then please provide secondary axis. As is, the figure will be incorrectly interpreted.

**Responses:** Thank you for this comment. We take your advice and uniform the scale. Please see
details in the revised Fig. 4.

Figure 1. Please provide ref. MA18 summer temperature reconstruction for comparison and discuss
it. This is a GDD >5 °C reconstruction; not sure why this is not presented and could not find
explanation in text.

**Responses:** Thank you. We have added the GDD₅ curve in the revised Fig. 1, and the corresponding
discussion in line 98-101 in the revised manuscript.

*“Regarding seasonal temperatures, the timing of the mid-Holocene HTM is in good agreement with*
*the multiproxy records from Kaufman et al.³ for both summer and winter seasons (KA20, Fig. 1b, c),*
*and also generally consistent with a summer-related record of growing degree days above 5 °C*
*(GDD₅) from North America and Europe⁹ (MA18, Fig. 1b).”*

Response to Reviewers

REVIEWERS' COMMENTS

Reviewer #3 (Remarks to the Author):

Review of the authors' response to original reviewers' comments

I am delighted to read the other two reviewers' positive and constructive comments on this excellent contribution, and the authors' meticulous, well-referenced responses accompanied with multiple re-analyses and re-considerations of the pre-treatment, analytical and presenting approaches.

The authors' demonstrate excellent understanding of the state of the art data, methods, and influence of approaches and tests regarding Holocene large-scale temperature reconstructions. What is more, they are clearly committed to look further and meticulously respond to the reviewers' suggestions and make careful changes to the text, analyses, tables, figures and supplementary materials.

The revision significantly strengthens the study, which, as I have previously stated is based on an impressive dataset, including much new data in particular from East Asia, a sound continental-only view, based on one climatically very sound proxy, i.e. pollen, which is then treated carefully through biomization and plant functional types PFTs. Thus it is a very welcome addition contributing to the topical discussion on Holocene climate trends, climate forcings and system feedbacks.

I went through all the authors' responses to all reviewer comments in detail and found them very thorough, I have nothing more to request. In some cases, where the reviewers' suggestions/thoughts were rebutted, the reasons were not only discussed in detail but also tested or visualized. I applaud the authors' for their careful and thorough responses.

I highlight the important changes, or improvements, made to the manuscript based on reviewers' comments:

- Figure 1 and discussion based on it is amended by adding presentation of geographic differences,
other small adjustments such as scaling, referencing, removal of IND record; this all clarify the
points discussed now in depth in the text

- Extensive testing and exploration of methodologies, including: significance testing of individual
reconstructions (now adjusted to 84.2% significance following Marsicek et al.), reconstructions
based on individual pollen taxa vs. PFTs, testing of other reconstruction methods (WA-PLS and
BRTs), testing of summer season variable of choice (GDD5 vs. warmest month of the year MTWA),
testing of spatial autocorrelation (changes in H-block distances), correction of isostatic rebound on
temperatures. All of these tests support the authors' prior judgement, support their choice of data
analyses, conclusions and importantly, render the results robust and thoroughly assessed. Most of the
key testing results are now presented in the supplementary materials.

- The authors now discuss their key results and their implications in much more detail, including
minor clarifications to sentences. There are major improvements to the abstract (key results), reasons
for year round warm mid-Holocene and late Holocene winter cooling (decoupled from orbital
forcing), the case of warm season bias in proxy reconstructions, and careful discussion of
significance testing and choice of methodologies

**Responses:** Thank you very much for your positive and valuable comments. We have carefully
checked the manuscript and prepared the materials complied with the editors' requirements.
